# HIV-1 treatment timing shapes the human intestinal memory B-cell repertoire to commensal bacteria

Cyril Planchais[1], Luis M. Molinos-Albert [1,7], Pierre Rosenbaum [1], Thierry Hieu [1], Alexia Kanyavuz[2], Dominique Clermont [3], Thierry Prazuck[4], Laurent Lefrou[5], Jordan D. Dimitrov [2], Sophie Hüe [6], Laurent Hocqueloux [4] & Hugo Mouquet [1] ✉

HIV-1 infection causes severe alterations of gut mucosa, microbiota and immune system, which can be curbed by early antiretroviral therapy. Here, we investigate how treatment timing affects intestinal memory B-cell and plasmablast repertoires of HIV-1-infected humans. We show that only class-switched memory B cells markedly differ between subjects treated during the acute and chronic phases of infection. Intestinal memory B-cell monoclonal antibodies show more prevalent polyreactive and commensal bacteria-reactive clones in late- compared to early-treated individuals. Mirroring this, serum IgA polyreactivity and commensal-reactivity are strongly increased in late-treated individuals and correlate with intestinal permeability and systemic inflammatory markers. Polyreactive blood IgA memory B cells, many of which egressed from the gut, are also substantially enriched in late-treated individuals. Our data establish gut and systemic B-cell polyreactivity to commensal bacteria as hallmarks of chronic HIV-1 infection and suggest that initiating treatment early may limit intestinal B-cell abnormalities compromising HIV-1 humoral response.

Antibodies are paramount in mucosal tissues as protective immune effectors against invading pathogens and immunomodulators of the microbiota[1]. Mucosal antibodies produced in the gut arise from either T-cell independent processes in the lamina propria and isolated lymphoid follicles, or germinal center (GC) reactions in lymphoid follicles of Peyer's patches and mesenteric lymph nodes[1–4]. According to the classical view, GC-induced high-affinity class-switched immunoglobulins respond to microbial threats, while T-cell independent-derived antibodies coat commensal bacteria[1]. Plasma cell and memory B-cell repertoires in intestinal mucosa are shaped by microbial exposures[5]. Microbiota changes also favor memory B-cell diversification that

generates persisting clonotype variants, while infection can trigger the recruitment of new intestinal B-cell clones[6]. Mucosal humoral immunity at different anatomical sites of the human body can involve IgM, IgD, IgG and/or IgA antibodies, the latter playing a predominant role for ensuring host-microbiota homeostasis in the gut[1,7,8]. Mucosal IgAs are mainly produced as J-chain-linked dimers (dIgA) and can form secretory IgAs (SIgA) when combined with the extracellular segment of the polymeric Ig receptor (pIgR) cleaved after IgA binding, named the secretory component (SC). Diverse mechanisms contribute to the antimicrobial functions of SIgA such as immune exclusion, adherence inhibition, agglutination, neutralization and enchained growth[8,9]. In

[1]Humoral Immunology Unit, Institut Pasteur, Université Paris Cité, INSERM U1222, F-75015 Paris, France. [2]Centre de Recherche des Cordeliers, INSERM, Sorbonne Université, Université de Paris, 75006 Paris, France. [3]Collection of the Institut Pasteur, Institut Pasteur, Université Paris Cité, 75015 Paris, France. [4]Service des Maladies Infectieuses et Tropicales, CHR d'Orléans-La Source, 45067 Orléans, France. [5]Service d'Hépato-Gastro-Entérologie, CHR d'Orléans-La Source, 45067 Orléans, France. [6]INSERM U955—Équipe 16, Université Paris-Est Créteil, Faculté de Médecine, 94000 Créteil, France. [7]Present address: ISGlobal, Hospital Clínic-Universitat de Barcelona, 08036 Barcelona, Spain. ✉e-mail: hugo.mouquet@pasteur.fr

addition, the promiscuous binding property of certain commensal-coating IgAs, known as polyreactivity, may account for their capacity to cross-react with commensal bacteria from different microbial taxa[10,11]. Indeed, a fraction of intestinal IgA and IgG plasmablast antibodies in humans show polyreactivity[12].

HIV-1 infection in mucosal tissues triggers a multifaceted cascade of pathological events affecting the integrity and function of the epithelial barrier, lymphoid structures and microbiota[13,14]. In the gut and associated lymphoid tissues, this involves epithelium destruction, bacterial translocation, dysbiosis, drastic depletion of CD4$^+$ T cells i.e. follicular helper T (Tfh) cells, loss of GC architecture and function, immune cellular metabolic reprogramming and inflammation[15–19]. HIV-1-induced GC disruption and subsequent impaired antibody responses may further favor the translocation of invading microbial products, chronic activation−exhaustion of immune cells, and biased or ineffective antibody responses[20–22]. Impairment in establishing optimal HIV-1 humoral responses combined with a potential immune diversion to translocating bacteria[23–25], could also compromise the local production of high-affinity antibodies targeting the viral envelope glycoproteins (Env). Still, ART initiated early during primary infection partially prevents HIV-1-induced mucosal damages and immune dysregulation[26,27]. Early ART allows the preservation of functional gut Tfh and Env-specific memory B cells[28–30], which may aid in developing functionally-relevant HIV-1 antibodies. Yet, the impact of ART timing on the microbiota composition and microbial translocation remains unclear[30–32]. Moreover, whether ART timing influences gut mucosal plasma cell and memory B-cell antibody repertoires and reactivities is still unknown.

Here, we investigated the impact of ART initiation timing on the intestinal B-cell antibody repertoires by studying IgG$^+$ and IgA$^+$ memory B cells and plasma cells from rectosigmoid colon tissues of individuals treated during the acute (eART) and chronic (lART) phases of infection. lART and eART harbored distinct intestinal memory B-cell antibody gene and reactivity repertoires. By characterizing a panel of 200 mucosal memory monoclonal antibodies, we found that poly- and commensals-reactive B-cell clones accumulate in the gut of lART. lART also exhibited augmented anti-commensals IgA seroreactivity, levels of systemic inflammation and bacterial translocation markers, and mucosa-derived polyreactive blood memory B-cell frequencies compared to eART. Together, these data suggest that HIV-1-induced gut mucosal damages and subsequent microbial translocation favor the elicitation of polyreactive intestinal memory B cells including clones circulating in the blood, which can be contained by early ART.

## Results

### ART timing tunes intestinal memory B-cell repertoires of HIV-1-infected individuals

To examine the effect of ART timing on intestinal B-cell repertoires, we first obtained colorectal biopsies from six HIV-1-infected individuals treated either during the acute or chronic phase of infection ($n = 3$ per group: eART, E1–E3 and lART, L1–L3) (Supplementary Table 1). All donors had serum IgG and IgA antibodies against HIV-1 Env proteins gp140, gp120 and gp41 (Fig. 1a and Supplementary Fig. 1a). Mucosal CD19$^+$ cells were mainly composed of CD27$^+$ memory B cells and antibody-secreting cells (CD27$^+$ CD38$^+$; ASC) expressing in majority surface IgA (36% vs. 22% for IgG$^+$ and 81% vs. 4% for IgG$^+$, respectively) (Fig. 1b and Supplementary Fig. 1b–d), as previously reported[25,28]. Of note, however, IgG$^+$ ASC frequencies may be underestimated by the cell surface staining as compared to an intracellular detection[12]. Flow cytometric analysis of mucosal B-cell subsets also showed a trend toward a decreased frequency of mature naïve (MN) cells, and conversely, an increased frequency of resting memory (RM) B cells in eART as compared to lART (Fig. 1b and Supplementary Fig. 1b–d) as previously reported[28]. Single mucosal class-switched (CS) IgA$^+$ and IgG$^+$ B cells and ASC were FACS-sorted (Fig. 1b and Supplementary Fig. 1b),

and their immunoglobulin gene sequences analyzed. No major alterations in antibody gene features, including somatic hypermutation, were observed for ASC between eART and lART, while significant differences were noticed for CS memory B cells (Fig. 1c and Supplementary Fig. 2a, b, f). As compared to eART, lART had CS B cells with increased frequencies of V$_H$3-Vλ3 associations ($p = 0.019$), and Vκ4-Jκ2 gene rearrangements ($p = 0.044$) (Fig. 1c, d and Supplementary Fig. 2c). In contrast, usage frequencies of Vλ2, Jκ4 and rearranged Vλ2-Jλ2/3 genes, as well as IgA/IgG ratio, were significantly reduced in lART compared to eART (24% vs. 44%, $p = 0.02$; 5% vs. 23%, $p = 0.047$; 1.3 vs. 2.4, $p = 0.017$; 10.6% vs. 14.9%, $p = 0.04$, respectively) (Fig. 1c and Supplementary Fig. 2b, c). Of note, anti-pneumococcal capsular polysaccharide-related antibody gene Vλ1-51[33] was more represented in lART CS B cells, albeit not statistically significant ($p = 0.07$; Supplementary Fig. 2f). Increased frequencies of positively charged CDRλ3 ($p = 0.016$) and neutral CDRκ3 ($p = 0.04$) were also found in lART CS B cells (Fig. 1c, d and Supplementary Fig. 2b). Moreover, somatic mutations in V$_H$ and Vκ genes were increased in CS B cells from lART compared to eART (19.6 vs. 17.5, $p = 0.037$; Supplementary Fig. 2g and 15 vs. 12.8, $p = 0.042$; Fig. 1e). Strikingly, higher levels of V$_H$ somatic mutation in lART were observed for IgA- but not IgG-expressing B cells (20.2% vs. 17.3% for IgA, $p = 0.013$ and 19.2 vs. 20.5 for IgG, $p = 0.39$; Fig. 1e).

Next, we carried out immunoglobulin high-throughput deep sequencing (Ig-HTS) on crude Ig DNA libraries (total B-cell and ASC populations) of matched colorectal and peripheral blood mononuclear cell (PBMC) samples from E1–E3 eART and L1–L3 lART donors plus four extra-donors per group. In total, 12,265,476 IgA and 4,669,125 IgG sequences from eART ($n = 7$; E1–E7) and 12,779,331 IgA and 7,300,158 IgG sequences from lART ($n = 7$; L1–L7) were analyzed. In line with immunoglobulin sequencing data on single memory B cells, lART showed a modest but significant increased number of somatic mutations in intestinal IgA-V$_H$ sequences (21.4 vs. 20.5, $p < 0.0001$) (Fig. 1f), and conversely, a very slight decrease of the IgG-V$_H$ mutation level compared to eART (21.9 vs. 22, $p < 0.0001$) (Fig. 1f). These minored differences compared to single memory B-cell data may be explained by the abundance in the gut libraries of Ig transcripts/amplicons derived from ASC, which showed opposite tendencies on the mutation loads between groups at the single-cell level (Supplementary Fig. 2g). The increased IgA-V$_H$ hypermutation in lART was not restricted to a single V$_H$ gene family (Fig. 1g), and highly mutated antibody sequences were found in both, narrowed and broad clonal expansions (Fig. 1h). Similarly, Ig-HTS data from PBMC libraries showed a higher averaged number of mutations in IgA-V$_H$ sequences in lART compared to eART (19.8 vs. 18.4, $p < 0.0001$) and in contrast, a lower number of IgG-V$_H$ gene mutations (19.8 for eART vs. 18.9 for lART, $p < 0.0001$) (Fig. 1f).

### Delayed ART in HIV-1-infected individuals promotes poly-reactive and self-reactive intestinal memory B cells

A substantial fraction of human intestinal IgA- and IgG-secreting cells exhibit natural polyreactivity with an innate specificity to microbiota[10,12]. We thus wondered whether variations in the intestinal B-cell repertoires of eART and lART could translate into distinct antibody reactivities. To address this, we produced a total of 200 randomly selected unique monoclonal antibodies (mAbs) from intestinal IgA$^+$ and IgG$^+$ memory B cells of six ART-treated HIV-1-infected individuals (3 eART E1–E3, $n = 101$; 3 lART L1–L3, $n = 99$) (Supplementary Tables 1 and 2). All antibodies were expressed as recombinant human IgG1 monoclonals to palliate potential variability effects in binding comparison due to the antibody format. To first check for high-affinity HIV-1 antibodies, we evaluated their binding to YU-2 trimeric gp140 and HXB2 p24 proteins by ELISA in stringent conditions. None of the 200 mAbs specifically recognized either gp140 or p24, but two of them (E2-136 and L3-127) displayed equivalent reactivities to both HIV-1 antigens (Fig. 2a and Supplementary Fig. 3a), suggesting a strong non-

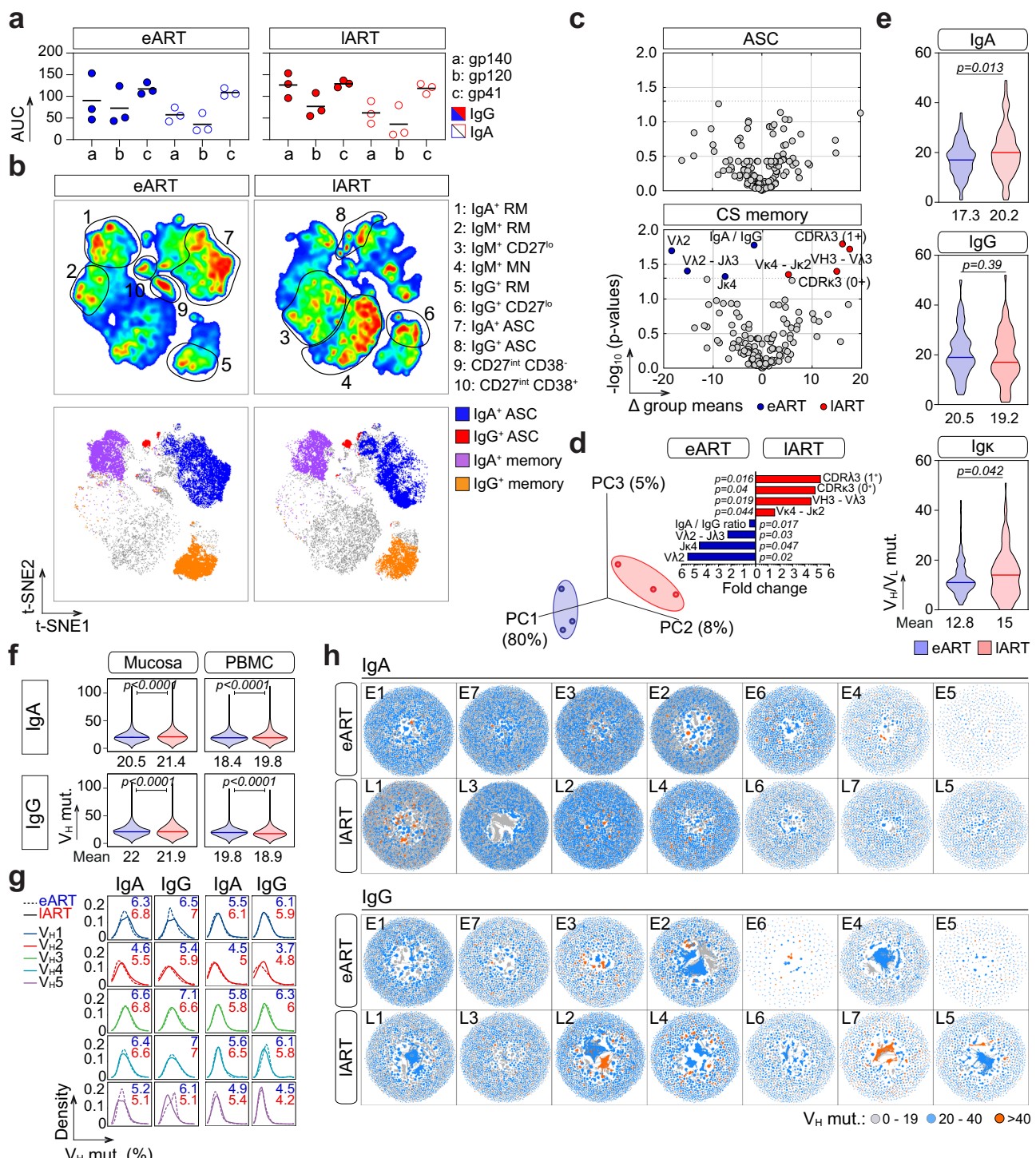

specific binding possibly due to polyreactivity. We thus determined the level of polyreactivity in the mAb panel against HIV-1 Env and unrelated antigens (i.e., keyhole limpet hemocyanin (KLH), double stranded (ds)DNA, Insulin and lipopolysaccharide (LPS)) by polyreactivity ELISA[34] (Fig. 2b, c). Antibodies, including E2-136 and L3-127, equally reacted against HIV-1 gp140, gp120, and gp41 proteins with a binding pattern reflecting polyreactivity (Fig. 2b and Supplementary Fig. 3b). The frequency of polyreactive antibodies tested against non-HIV-1 antigens varied between individuals ([7.4–25.7%] for eART and [25–38.9%] for lART), but was on average statistically higher in lART compared to eART (31.3% vs. 18.8, $p = 0.041$) (Fig. 2c, e and Supplementary Fig. 3c). The more elevated global level of polyreactivity in

lART was mainly attributed to IgA-expressing B cells (35% vs. 19%, $p = 0.033$ for IgA[+]; 28% vs. 19%, $p = 0.36$ for IgG[+]) (Fig. 2e). Polyreactive antibodies showed increased frequencies of $V_H4$ ($p = 0.046$), $J_H2$ ($p = 0.05$), rearranged Vλ1-Jλ1 ($p = 0.03$) and Vκ3-Jκ2 ($p = 0.05$) genes, and reciprocally a decreased frequency of Vκ1-Jκ2 ($p = 0.035$) compared to non-polyreactive antibodies (Supplementary Fig. 3d, f). Polyreactive antibodies also had higher Vκ mutation rates (14.6% vs. 11.2% for non-polyreactive mAbs, $p = 0.01$), and possessed more negatively charged $CDR_H3$ loops (92% vs. 39% for non-polyreactive mAbs, $p = 0.014$) (Supplementary Fig. 3d–f). Conversely, compared to polyreactive antibodies, non-polyreactive mAbs displayed an increased frequency of positively charged CDRλ3 (47% vs. 13% for

**Fig. 1 | Immunoglobulin gene repertoire of intestinal B cells from eART and lART. a** Dot plots comparing the serum anti-HIV-1 Env IgG (colored) and IgA (clear) antibody levels in eART (blue) and lART (red) individuals ($n = 3$ per group). The $y$ axis indicates the area under the curve (AUC) values of the ELISA binding curves shown in Supplementary Fig. 1a. Bars correspond to the means. Samples were tested in two independent experiments. **b** t-SNE-based analysis comparing the subset distribution of single mucosal CD19+ cells between eART and lART donors ($n = 3$; $2 \times 10^5$ cells per group) (top). Single-cell sorted B-cell sub-populations are shown in t-SNE plots (bottom). **c** Volcano plots comparing the immunoglobulin gene repertoires of intestinal antibody-secreting cells (ASC) and class-switched (CS) memory B cells ($n = 206$ parameters) between e-ART (blue) and l-ART (red). Dashed lines indicate the statistically significant cut-off ($p < 0.05$). **d** Plot showing the principal component analysis (PCA) of intestinal CS B cells in eART and lART (left). Contribution plot showing the fold changes of significantly diverging parameters between groups (right). Groups in (**c**) and (**d**) were compared using $2 \times 2$ Fisher's Exact test. **e** Violin plots comparing the number of somatic mutations in the IgA/IgG $V_H$ ($n = 143$ for eART and $n = 138$ for lART) and Vκ genes ($n = 96$ for eART

and $n = 88$ for lART) from single-sorted intestinal CS B cells between eART ($n = 3$) and lART ($n = 3$) donors. **f** Violin plots comparing the number of somatic mutations in the IgA ($n = 12,265,476$ and 12,779,331 sequences for eART and lART, respectively) and IgG $V_H$ ($n = 4,669,125$ and 7,300,158 sequences for eART and lART, respectively) genes from mucosal and peripheral blood B cells analyzed by NGS between eART ($n = 7$) and lART ($n = 7$). The average number of mutations is indicated below each violin plot. Numbers of hypermutation in (**e**) and (**f**) were compared between groups using two-tailed unpaired Student's $t$ test with Welch's correction. Bars in (**e**) and (**f**) represent the medians. **g** Divergence plots comparing the distribution of intestinal IgA+ and IgG+ B-cell sequences ($y$-axis) between eART (dashed line) and lART (straight line) according to their $V_H$-gene family and hypermutation frequencies ($x$-axis). The average frequencies of mutations for eART (blue) and lART (red) are indicated in each plot. **h** Network visualization comparing the clonal expansion levels of intestinal of IgA+ and IgG+ B cells according to the somatic mutation loads between eART and lART HIV-1-infected individuals. Source data are provided as a Source Data file.

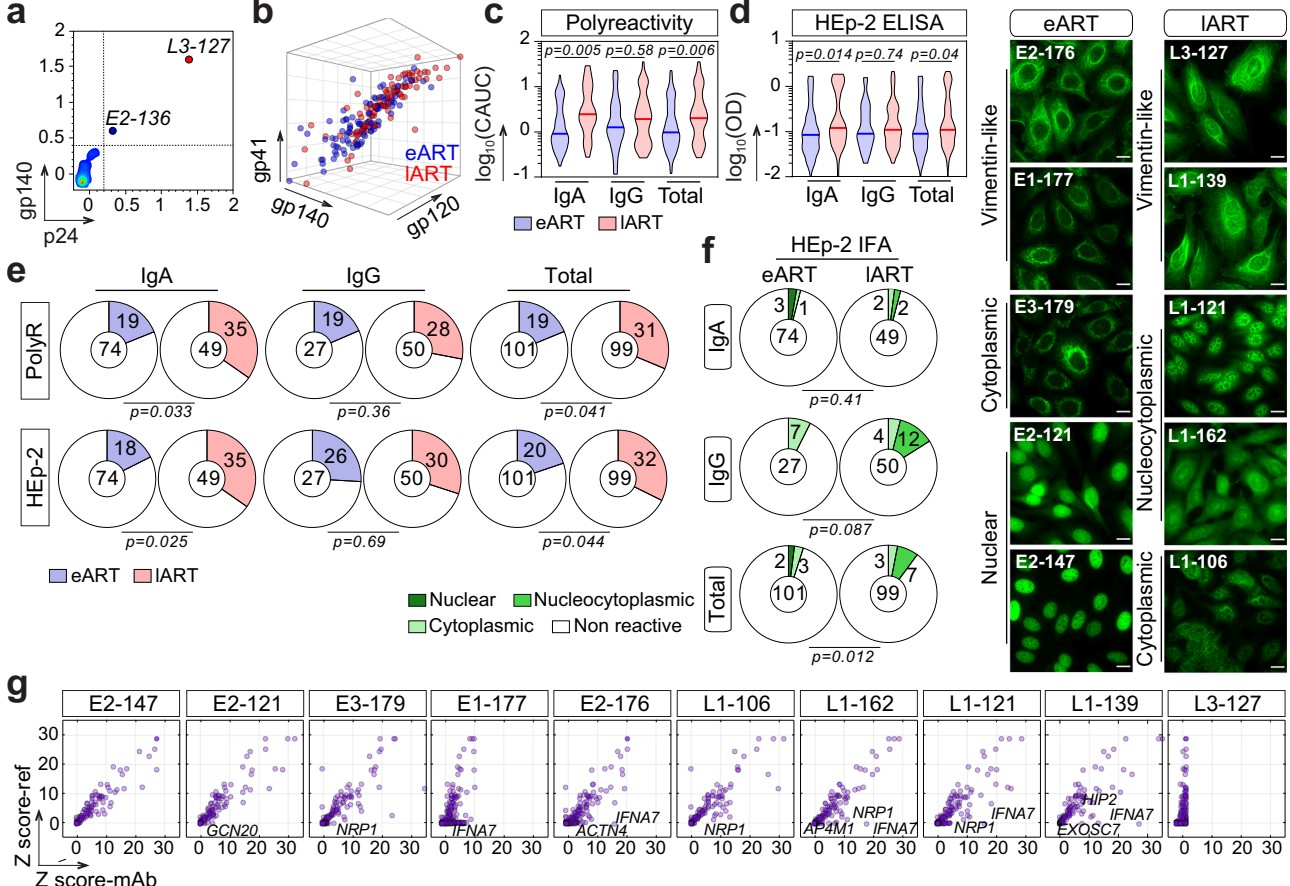

**Fig. 2 | Poly- and self-reactivity of intestinal memory B-cell antibodies from eART and lART. a** Plot comparing the ELISA reactivity of mucosal CS memory B-cell antibodies against HIV-1 Env gp140-F vs. p24 ($n = 200$). Antibodies were tested in triplicate. **b** 3D dot plot comparing the polyreactive binding of mucosal CS memory B-cell antibodies from eART (blue) and lART (red) to HIV-1 proteins as measured by ELISA in Supplementary Fig. 3a. **c** Violin plots comparing the ELISA polyreactivity of mucosal CS memory B-cell antibodies from eART (blue) and lART (red). The $y$-axes indicate the $\log_{10}$ cumulative AUC (CAUC) values for polyreactivity as measured in Supplementary Fig. 3c. Antibodies were tested in two independent experiments. Groups were compared using two-tailed Student's $t$ test with Welch's correction. **d** Same as in (**c**) but for HEp-2-reactivity with ELISA $OD_{405\,nm}$ values for the binding to HEp-2 cell antigens. Bars in (**c**) and (**d**) represent the medians. Antibodies were tested in triplicates. **e** Pie charts comparing the frequency of poly- and self-reactive of intestinal IgA+ and IgG+ CS B-cell antibodies between eART (blue) and lART (red)

as measured in (**c**) and (**d**), respectively. The number of tested antibodies is indicated in the pie chart center, and the frequency of the reactive ones (colored) on the chart. Groups were compared using $2 \times 2$ Fisher's Exact test. **f** Microscopic images showing representative antibody reactivities to HEp2-expressing self-antigens detected by indirect immunofluorescence assay (IFA) (representative from two independent experiments). Scale bars represent 15 μm. Pie charts summarizing the IFA data are shown. The number of tested antibodies is indicated in the pie chart center, and the frequency of the reactive ones (colored) on the chart. Groups were compared using $2 \times 5$ Fisher's Exact test. **g** Microarray plots showing the reactivity profile of selected antibodies to human proteins. For each protein spot, $Z$-scores given by the reference (Ref: mGO53) and test antibody are depicted on the $y$- and $x$-axis, respectively. Immunoreactive proteins are indicated in each plot. Source data are provided as a Source Data file.

polyreactive mAbs, $p = 0.02$), shorter CDRκ3 ($p = 0.02$), and a tendency for increased $D_H4$ gene frequency ($p = 0.05$) (Supplementary Fig. 3d, f). Of note, $V_H4$-34 and $V_H4$-59 genes previously associated with antibody polyreactivity[35,36], were more prevalent among polyreactive mAbs compared to their non-polyreactive counterparts (8% vs. 0.67% for $V_H4$-34, $p = 0.014$, and 14% vs. 7.3% for $V_H4$-59, $p = 0.049$, respectively) (Supplementary Fig. 3g).

Human blood IgG[+] but not IgA[+] memory B-cell antibodies are often self-reactive[37,38]. By contrast, bone marrow plasma cells and intestinal plasmablasts rarely produce genuine self-reactive IgG antibodies[10,12,39]. To evaluate the self-reactivity of intestinal CS B-cell antibodies from ART-treated individuals, we first measured their ELISA binding to HEp-2 cell antigens (Fig. 2d). The frequency of self-reactive antibodies varied between donors ([8–27%] for e-ART and [28–40%] for lART) and was on average statistically higher for lART compared to eART (32% vs. 20%, $p = 0.044$) (Fig. 2e). In agreement with polyreactivity data, the more elevated global level of HEp-2 reactivity in lART compared to eART was mainly attributed to IgA-expressing B cells (35% vs. 18%, $p = 0.035$ for IgA[+] and 30% vs. 26%, $p = 0.79$ for IgG[+]) (Fig. 2d, e). Both poly- and HEp-2-reactivity were correlated ($r = 0.7698$; $p < 0.0001$; Supplementary Fig. 3h). We next assayed the HEp-2 cell-reactivity of the mAbs by indirect immunofluorescence assay (IFA), and classified the staining patterns as either nuclear, cytoplasmic or nucleocytoplasmic (Fig. 2f). IFA HEp-2-reactivity frequency ranged from 3 to 7.1% for e-ART and 7.1 to 11.4% for lART, and was on average significantly increased in lART compared to eART (10.1% vs. 5%, $p = 0.012$) (Fig. 2f). However, in contrast to ELISA binding data, IFA HEp-2-reactivity of lART antibodies mainly originated from IgG-expressing B cells (IgA[+]: 4% vs. 4% for eART, $p = 0.41$; IgG[+]: 16% vs. 7% for eART, $p = 0.087$) (Fig. 2f). The reactivity of ten selected HEp-2 cell binders ($n = 5$ per group) was then probed by microarray immunoblotting against ~9500 unique human proteins (Fig. 2g and Supplementary Fig. 3i). About a third of the antibodies, all giving anti-vimentin-like IFA staining patterns (E1-177, E2-176 and L3-127), showed polyreactive binding on the protein arrays (Supplementary Fig. 3i). Of note, these antibodies and two nucleus/cytoplasm-reactive mAbs (L1-162 and L1-121) cross-reacted strongly with interferon alpha-7 protein (IFNA7; Fig. 2g). Highly significant immunoreactive protein hits ($Z$ score >5) were comparably detected for both eART- and lART-derived intestinal mAbs (Fig. 2g).

## Commensals-reactive intestinal memory B cells and circulating blood IgA antibodies accumulate in late-treated HIV-1-infected individuals

A prime function of the gut polyreactive antibodies is to prevent bacterial translocation and to regulate the microbiota composition[7,8,11,40]. To examine the impact of ART timing delay on the mucosal B-cell reactivity against commensals, we evaluated the ELISA binding of e-ART and l-ART mucosal mAbs against seven bacteria strains composing the intestinal microbiota: *Escherichia coli*, *Enterococcus faecalis*, *Enterobacter cloacae*, *Morganella morganii*, *Bacillus subtilis*, *Clostridium difficile* and *Prevotella corporis*. Compared to eART, lART-derived antibodies as a group were significantly more reactive against all the bacteria ($p$ range [0.026–0.001]; Fig. 3a). Of note, bacteria and polyreactive bindings of the 200 mAbs were strongly correlated (Fig. 3b). In this collection of intestinal antibodies, we identified one non-polyreactive mAb from a lART individual, L2-168, binding specifically to *B. subtilis* and not to other *Bacilliaceae* strains (Fig. 3a–c). To further characterize the antigen recognized by L2-168, we first examined its reactivity against whole-fixed *B. subtilis* bacteria by IFA, which showed a fine cell-wall staining pattern (Fig. 3d). *Bacillus subtilis* is a Gram-positive bacteria whose cell wall is composed of a thick peptidoglycan layer mainly containing teichoic acids, lipoteichoic acids (LTA) and flagellin[41]. We found that L2-168 bound to *B. subtilis* LTA but not peptidoglycan and flagellin (Fig. 3e, f), and did not cross-react against LTA from *Staphylococcus aureus* or *Streptococcus pyogenes* (Fig. 3e).

Peripheral blood B-cell subset and secreted antibody compartments are also subjected to major alterations in HIV-1-infected patients[42], which are partially reverted by early ART[43]. Thus, we next assayed whether ART timing influences the reactivity of serum antibodies to both HIV-1 and gut commensals using larger cohorts of eART ($n = 38$) and lART ($n = 40$) donors (Supplementary Table 1). All purified serum IgG and IgA antibodies reacted against HIV-1 gp140 protein albeit, at higher levels for IgGs than IgAs (Fig. 4a). Anti-Env IgG and IgA antibody titers, and IgG-mediated in vitro neutralizing activities were significantly greater for lART than e-ART (65% vs. 28.2%, $p = 0.0003$ for neutralization) (Fig. 4a, b and Supplementary Fig. 4a). No major differences were evidenced for IgA and IgG polyreactive bindings to non-HIV-1 antigens between both groups ($p = 0.94$ and $p = 0.53$, respectively) (Supplementary Fig. 4b). However, serum LPS-reactive IgAs but not IgGs were significantly enriched in lART compared to eART ($p = 0.03$ and $p = 0.26$, respectively) (Fig. 4c and Supplementary Fig. 4c). Humans possess two IgA subclasses, IgA1 and IgA2, and we found that IgA1 antibodies were mainly responsible for the increased anti-LPS IgA reactivity in lART ($p = 0.008$ vs. $p = 0.98$ for IgA2) (Fig. 4c). We next measured serum antibody levels against the aforementioned commensal bacteria strains ($n = 7$). Antibacterial IgG reactivities were globally comparable between eART and lART, although *E. faecalis*-binding IgG titers were more elevated in eART ($p = 0.0019$) (Fig. 4d). In contrast, serum IgAs from lART had increased reactivities to 4 out of the 7 commensals ($p = 0.017$, $p = 0.009$, $p = 0.01$ and $p = 0.009$ for *E. cloacae*, *B. subtilis*, *C. difficile* and *P. corporis*, respectively) (Fig. 4d). Moreover, serum IgA and IgG antibodies showed distinct global commensals-binding patterns (Supplementary Fig. 4d). Strikingly, bacteria-reactive intestinal mAbs and serum IgAs showed similar binding profiles, suggesting common recognition features between gut memory B-cell and circulating blood IgA antibodies (Fig. 4e).

## Increased systemic secretory IgA and inflammatory cytokine levels in late-treated HIV-1-infected individuals

Early after HIV-1 infection, the intestinal mucosa is subjected to intensive HIV-1 replication, massive CD4[+] T-cell depletion, intestinal epithelial barrier breakdown, and microbial translocation into the bloodstream leading to a systemic immune activation[30,44,45]. Efficient control of HIV-1 replication by ART significantly reduces the microbial translocation and chronic immune activation that, however, remains higher than in uninfected individuals[30]. To determine whether the delay of ART initiation could favor the translocation of mucosal IgAs into the circulation, we measured by ELISA the serum levels of dimeric and secretory IgA antibodies (dIgA and SIgA, respectively) in eART and lART. Total dIgA and SIgA titers were both significantly higher in lART than in eART ($p = 0.0004$ and $p < 0.0001$, respectively) (Fig. 5a), and were correlated ($r = 0.28$, $p = 0.0114$) (Fig. 5b). Serum SIgA titers correlated with the ELISA binding levels of serum IgAs to LPS, *B. subtilis*, *C. difficile* and *P. corporis* ($p = 0.0028$, $p = 0.0189$, $p = 0.0252$ and $p = 0.0085$, respectively), suggesting that blood mucosa-derived SIgA antibodies could contribute to the bacteria reactivity of total serum IgAs (Supplementary Fig. 4e, f). We then investigated whether delayed ART initiation contributes to sustained intestinal impermeability, which could explain greater amounts of SIgA in the blood and chronic systemic immune activation. We determined by ELISA the serum concentration of soluble CD14 (sCD14) and Reg3α, considered as indirect markers of gut permeability and bacterial translocation[30,46] (Fig. 5c). eART and lART showed similar levels of circulating sCD14 (1.8 μg/ml [1.4–3.6 μg/ml] vs. 1.9 μg/ml [1.1–3.9 μg/ml], respectively, $p = 0.29$) (Fig. 5c). However, the average concentration of serum Reg3α was heightened in lART compared to eART ($11.1 \pm 9.3$ ng/ml vs.

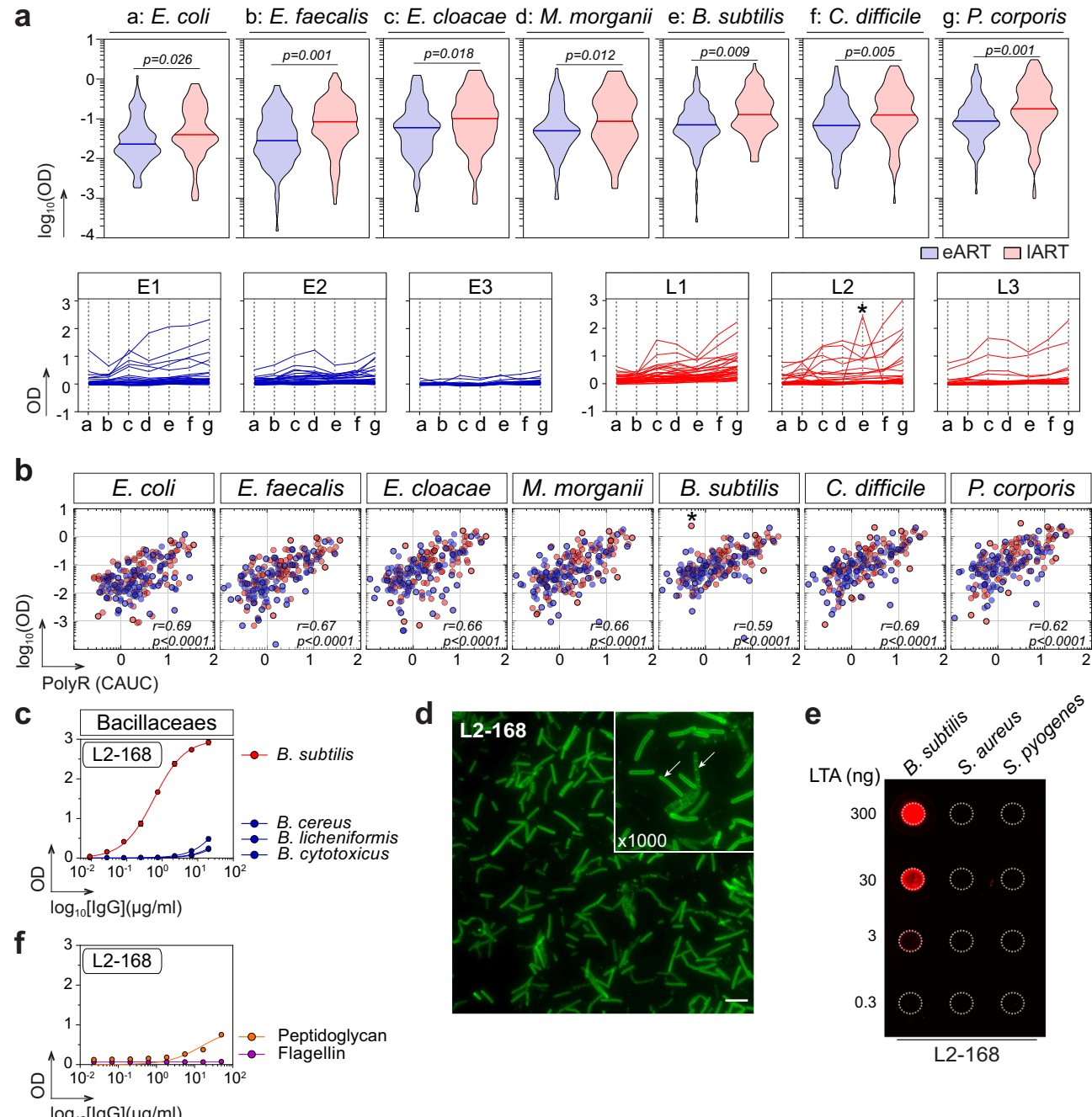

**Fig. 3 | Antimicrobial binding of intestinal memory B-cell antibodies from eART and lART. a** Violin plots (top) comparing the antibody binding to commensal bacteria between eART (blue) and lART (red) donors. Averaged values of antibodies (*n* = 200) tested in duplicate in two independent experiments are shown. Groups were compared using two-tailed unpaired Student's *t* test with Welch's correction. Plots (bottom) comparing the reactivity pattern of individual antibodies to commensal bacteria between eART (blue) and lART (red). * indicates that L2-168 antibody specifically binds to *B. subtilis*. Bars correspond to the medians. Antibodies were tested in duplicate in two independent experiments. **b** Correlation plots of polyreactivity vs. binding of intestinal memory B-cell antibodies to commensal bacteria. The *x*-axis indicates the polyreactivity CAUC values presented in Fig. 2b.

Bivariate correlations were estimated with the two-tailed Pearson correlation test. **c** Representative ELISA graphs showing the reactivity of L2-168 antibody against selected *Bacillaceaes* strains (*B. cereus*, *B. licheniformis*, *B. cytotoxicus* and *B. subtilis*). Means ± SEM of triplicate values are shown. **d** Microscopic image showing the IFA binding of L2-168 antibody to *B. subtilis* (Magnifications ×100 and ×1000) (representative of two independent experiments). The scale bar represents 15 μm. **e** Dot blot comparing the reactivity of L2-168 to purified LTA from Gram+ bacteria: *B. subtilis*, *S. aureus* and *S. pyogenes*. **f** Same as in (**c**) but for purified peptidoglycan and flagellin from *B. subtilis*. L2-168 antibody was tested in triplicate. Source data are provided as a Source Data file.

7.5 ± 4.6 ng/ml, respectively, *p* = 0.036) (Fig. 5c). The concentration of serum IL-8, which is directly linked to inflammation[47], was also significantly increased in lART (46.8 ± 73.2 pg/ml vs. 10.3 ± 16.7 ng/ml for eART, *p* = 0.0039), and correlated with the one of circulating sCD14 but not Reg3α in multivariate analyses combining clinico-virological and serological parameters (*p* = 0.0031 and *p* = 0.4686, respectively)

(Fig. 5c, d). In addition, serum SIgA levels correlated with the sCD14 and IL-8 concentrations (*p* = 0.035 and *p* = 0.0042, respectively) (Fig. 5d and Supplementary Fig. 4g). Multivariate analyses confirmed the significant association of SIgA titers with serum dIgA and anti-commensals IgA levels, bacteria-binding and polyreactivity of IgA and IgG antibodies (Fig. 5d). It also revealed strong positive correlations

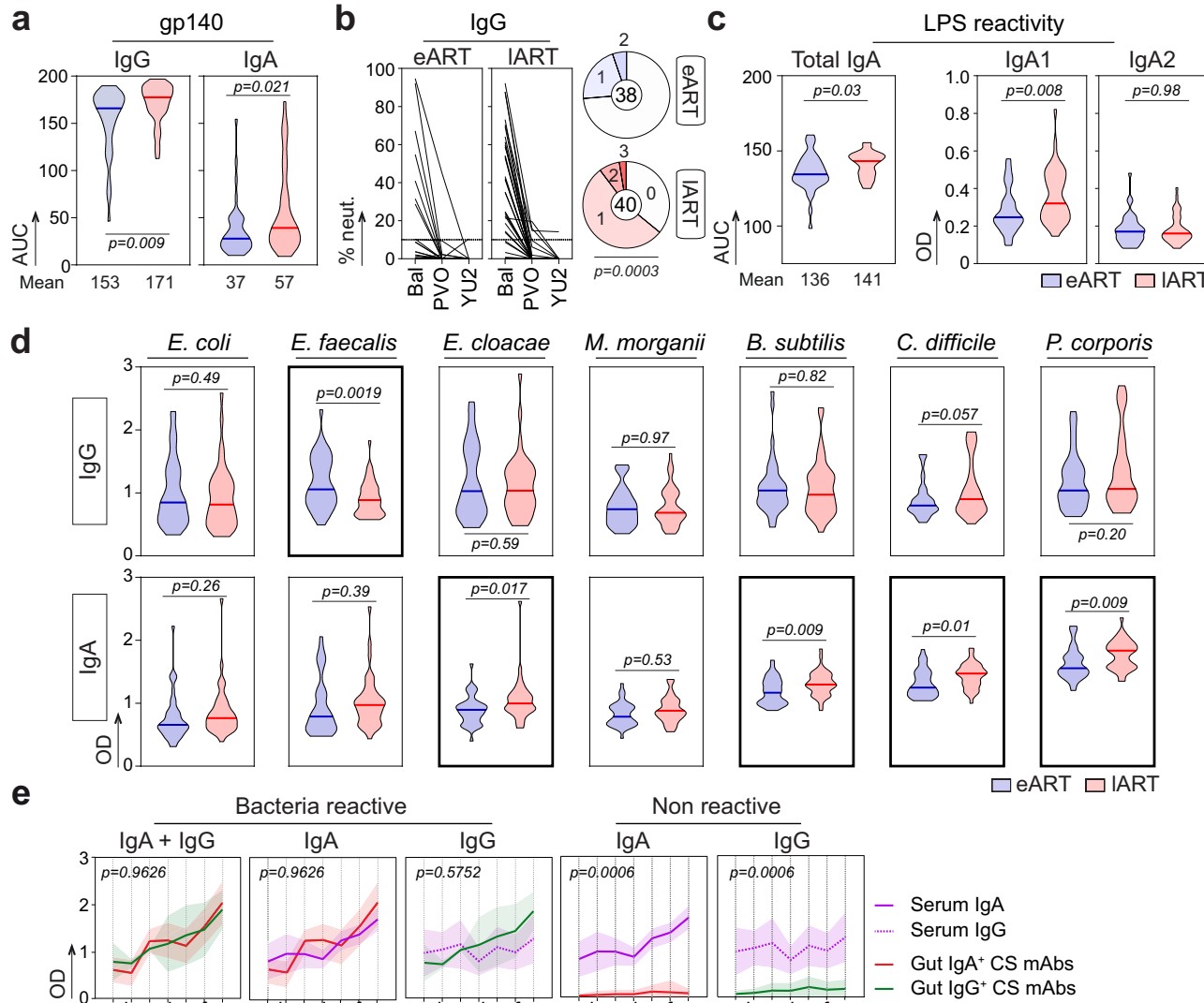

**Fig. 4 | Serum IgG and IgA antibody profiling in eART and lART. a** Violin plots comparing the binding of purified serum IgA and IgG antibodies to gp140 between eART (blue, $n = 38$) and lART (red, $n = 40$). The mean AUC is indicated below each violin plot. Bars represent the medians. Groups were compared using two-tailed unpaired Student's $t$ test with Welch's correction. Antibodies were tested in duplicate. **b** Graphs comparing the in vitro neutralization activity of purified serum IgGs against Bal.26 (Bal), YU2.DG (YU2) and PVO.4 (PVO) between eART ($n = 38$) and lART ($n = 40$). Means from triplicates from TZM-bl assay experiments are shown. Pie charts summarize the frequency of individuals with IgG-mediated neutralizing activities (>10%) against one (1), two (2) or three (3) viruses (colored). White color indicates that no neutralization was detected (0). Groups were compared using 2 × 5 Fisher's Exact test. **c** Violin plots comparing the binding of serum IgA, IgA1 and IgA2 antibodies to LPS between eART (blue, $n = 38$) and lART (red, $n = 40$). Means of triplicate values are shown. **d** Same as in (**c**) but for purified serum IgG and IgA antibodies against selected commensal bacteria. Bars in (**c**) and (**d**) represent the medians. Groups were compared using two-tailed unpaired Student's $t$ test with Welch's correction. Antibodies were tested in duplicate. **e** Graphs showing the reactivity profiles against commensal bacteria of purified serum antibodies (IgA, purple straight lines; IgG, purple dashed lines; $n = 78$ in total) and bacteria-reactive monoclonal antibodies (mucosal IgA, red line ($n = 10$) and mucosal IgG, green line ($n = 5$)). Data are presented as mean values ± SD. Cumulative distributions between groups were compared using the two-tailed Kolmogorov–Smirnov unpaired test. Source data are provided as a Source Data file.

between the time of infection without treatment and commensals-reactivities of serum IgA antibodies (Fig. 5d). Conversely, anti-HIV-1 Env and commensals-binding values were inversely correlated (Fig. 5d), suggesting that the viral persistence drives IgG antibody response to HIV-1 but not to commensal bacteria. Principal component analysis (PCA) with the same variables recapitulated a partial but marked segregation of eART and lART populations (Fig. 5e). Categorized parameters (as shown in Fig. 5d) were globally grouped by the dimensionality reduction (Fig. 5e). Among those, clinico-virological, inflammation and bacterial translocation parameters as well as anti-HIV-1-Env IgG/IgA and anti-commensals IgA titers allowed discriminating eART from lART, while anti-commensals IgG and

polyreactivity levels had a limited contribution to eART vs. lART clusterization (Fig. 5e).

**Gut-related polyreactive blood CD27⁻ IgA⁺ memory B cells are more prevalent in late-treated HIV-1-infected individuals**
A fraction of human blood IgAs and IgA-expressing B cells likely originate from mucosal tissues following the induction of a T-cell dependent or independent antigen response. IgA class-switched B cells generated by the latter express β7 integrin, lack CD27 surface markers, and are enriched in polyreactive clones recognizing various bacterial species[48,49]. Thus, we first examined by flow cytometric phenotyping whether circulating blood IgA⁺ and IgG⁺ memory B-cell

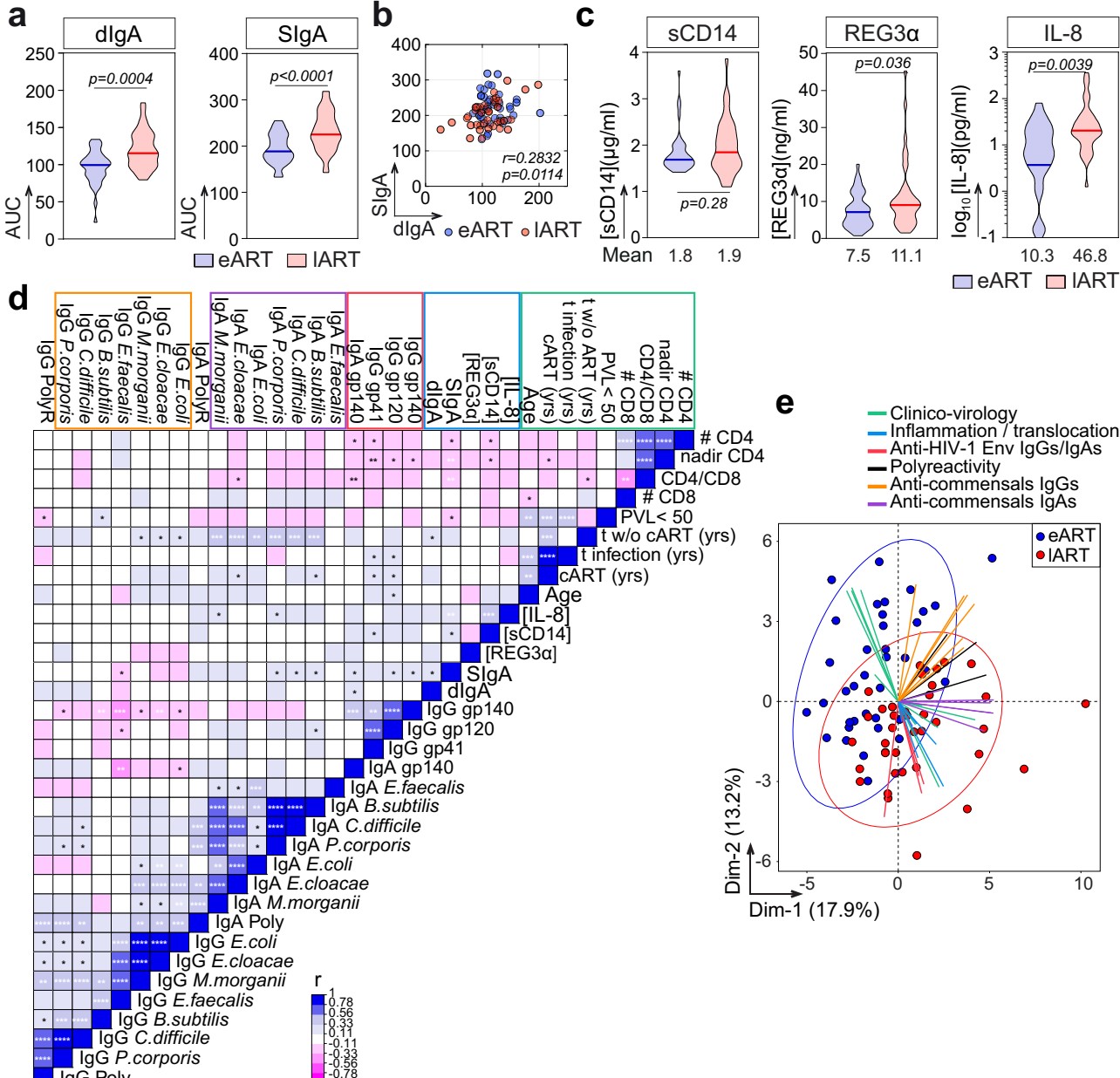

**Fig. 5 | Immuno-serological and clinico-virological parameters in eART and lART. a** Violin plots comparing the serum levels of dimeric (dIgA) (left) and secretory (SIgA) (right) IgA antibodies between eART (bleu) and lART (red). **b** Correlation plot comparing the levels of dIgA and SIgA determined by ELISA in eART (blue) and lART (red) (bottom). Bivariate correlation was estimated using the two-tailed Pearson correlation test. **c** Same as in (**a**) but for sCD14, REG3α, and IL8. The mean concentration from duplicate values is indicated below each violin plot. Bars in (**a**) and (**c**) represent the medians. Groups were compared using the two-tailed Student's *t* test with Welch's correction. **d** Correlation matrix of immunological and clinico-virological parameters (*n* = 34) from eART (*n* = 38) and lART individuals (*n* = 40). Cells are color-coded according to the value of two-sided Pearson rho correlation (*r*) coefficients. Asterisks correspond to unadjusted *p* values. ****p* < 0.0001, ***p* < 0.001, **p* < 0.01, *p* < 0.05. *p* values under the Benjamini–Hochberg-corrected significance threshold (*p* = 0.006) are highlighted in white. **e** PCA 2D-plot showing the clinico-virological and serological variables (color-coded) discriminating eART (blue, *n* = 38) and lART (red, *n* = 40). The two first dimensions account for 31.1% of the variability. Source data are provided as a Source Data file.

subsets differed according to treatment initiation timing (Fig. 6a). A reduced proportion of IgA⁺CD27⁺ B cells expressing β7 was evidenced in lART compared to eART, but did not reached statistical significance (10% vs. 17%, *p* = 0.06) (Fig. 6b and Supplementary Fig. 5b). Conversely, the proportion of IgG⁺CD27⁻ β7⁺ and β7⁻ B cells were significantly elevated in lART (15% vs. 6%, *p* = 0.032 and 25% vs. 10%, *p* = 0.016, respectively) (Fig. 6b and Supplementary Fig. 5b). Then, to determine the frequency of HIV-1 Env-reactive B cells in eART and lART, blood IgA⁺ and IgG⁺ B cells were stained with fluorescently labeled YU-2 gp140-F trimers (Fig. 6c). gp140⁺ class-switched B-cell frequencies

were on average higher in eART compared to lART as previously observed[28,50] (Fig. 6c, d). In both groups, gp140-reactive cells were more prevalent in the IgG⁺ than IgA⁺ memory B-cell pool (0.04% vs. 0.004% for eART, and 0.008% vs. 0.001% for lART, respectively) (Fig. 6d). In addition, a trend for a higher proportion of gp140-binding B cells expressing CD27 but not β7 surface marker was observed in eART (51% vs. 27% for lART) (Fig. 6c). Conversely, a substantial fraction of gp140-specific CS B cells in lART tended to have a CD27⁻ β7⁺ phenotype (10% vs. 28% for eART and lART, respectively) (Fig. 6c). Finally, to identify polyreactive clones among circulating B cells, we also

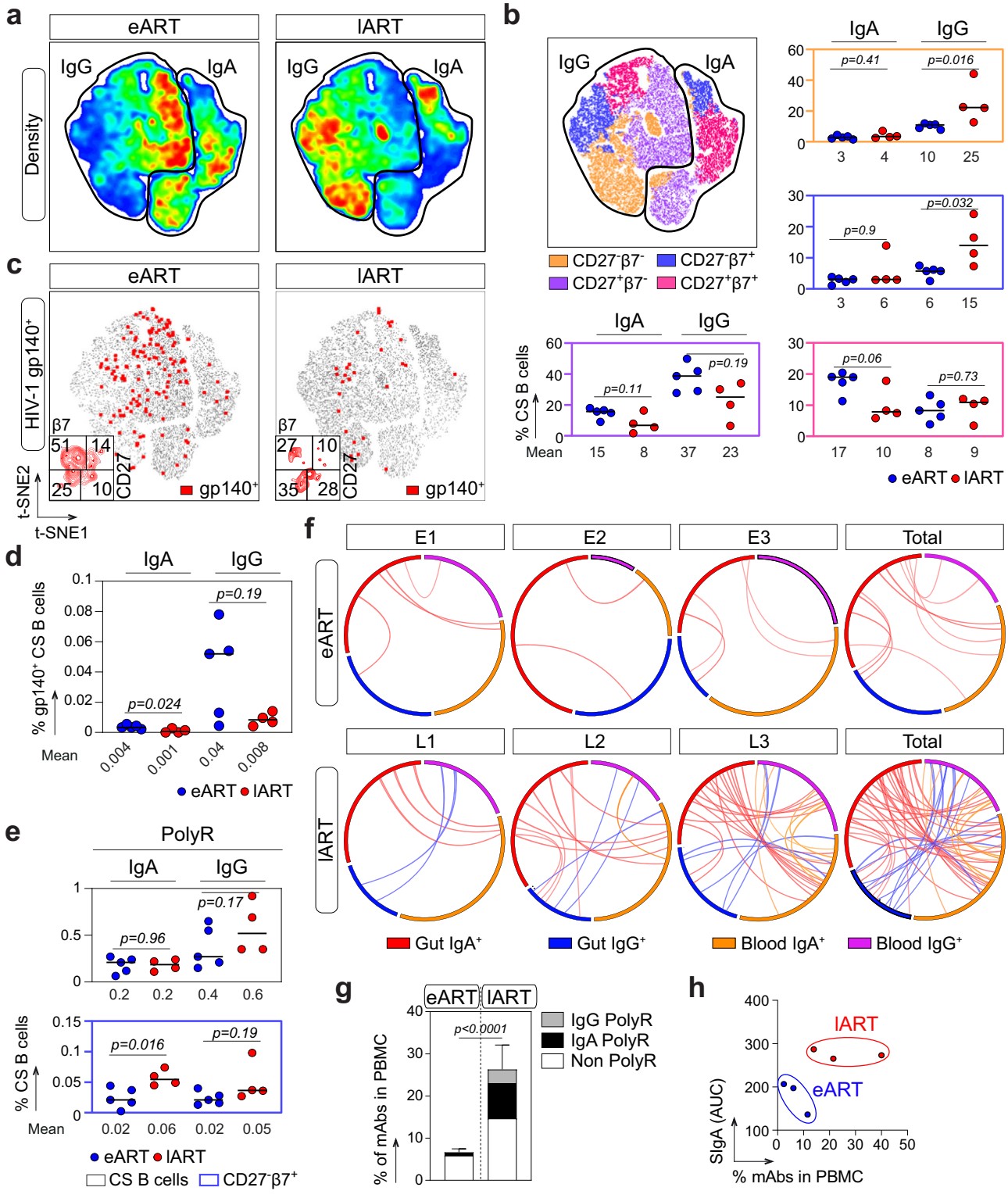

stained peripheral blood B lymphocytes with fluorescently labeled RNApol α, Insulin and LPS used as unrelated ligands, and determined the percentage of triple antigen-binding cells (PolyR+) by flow cytometry (Fig. 6e and Supplementary Fig. 5a). We found similar proportions of PolyR+ total blood IgA+ and IgG+ B cells in eART and lART (0.2% vs. 0.2% for IgA+, $p = 0.96$ and 0.4% vs. 0.6% for IgG+, $p = 0.17$, respectively) (Fig. 6e). Polyreactive clones were enriched in IgA+ CD27− B-cell subsets compared to CD27+ B cells for both eART and lART as previously reported in non-HIV-1-infected individuals[49], but the differences were not statistically significant ($p = 0.06$ for eART and $p = 0.2$

for lART) (Supplementary Fig. 5c). Importantly, however, polyreactive IgA+CD27−β7+ B cells were significantly enriched in lART compared to eART (0.06% vs. 0.02%, $p = 0.016$) (Fig. 6e and Supplementary Fig. 5c), despite comparable total IgA+CD27−β7+ B cell frequencies between groups (Fig. 6b).

To investigate whether ART timing modulates the proportion of intestinal B cells circulating in the blood, we searched for mucosal B-cell monoclonal IgA and IgG $V_H$ sequences in the Ig-HTS libraries from PBMC samples. We found much higher percentage of mucosal mAb sequences in the blood B cells of lART than eART (33.3% [33/99]

**Fig. 6 | Env- and poly-reactivity of blood memory B cells from eART and lART.** **a** Density plots showing the t-SNE analysis of circulating blood IgA[+] and IgG[+] CD19[+] cells from eART ($n = 5$) and lART ($n = 4$) ($2 \times 10^5$ cells per group). **b** t-SNE plot showing the distribution of blood B-cell subsets (based on CD27 and β7 markers) in circulating blood IgA[+] and IgG[+] CD19[+] cells from eART ($n = 5$) and lART ($n = 4$). Dot plots comparing the blood B-cell subset frequencies between eART (blue) and lART (red) are shown. The average frequency of positive cells is indicated below each dot plot. **c** t-SNE density plots comparing the distribution of HIV-1 gp140-specific cells (shown in red), with a phenotypic analysis of these cells based on CD27 and β7 surface expression at the bottom left-hand corner. **d** Dot plots comparing gp140[+] B-cell frequencies between eART ($n = 5$) and lART ($n = 4$). The average frequency of reactive cells among total IgA[+] and IgG[+] B cells is indicated below each dot plot. **e** Same as in (**b**) but for polyreactive B cells (PolyR[+]). Bars in (**b**), (**d**) and (**e**) represent the means. Groups were compared using two-tailed Mann–Whitney test. **f** Circos plots comparing the antibody sequence relationships between single-cell-sorted mucosal B cells and NGS libraries between cellular compartments. Inter-connecting lines indicate sequences sharing identical $V_H$ and $J_H$ gene segments and at least 90% $CDR_H3$ amino acid sequence homology. Groups were compared using $2 \times 2$ Fisher's Exact test. **g** Bar graph comparing the frequencies of polyreactive (gray and black for IgG and IgA, respectively) and non-polyreactive (white) intestinal memory B-cell antibody sequences found in the blood-derived NGS library between eART ($n = 3$) and lART ($n = 3$). Data are presented as mean values ± SD. Groups were compared using $2 \times 2$ Fisher's Exact test. **h** Dot plot presenting the clustering of eART (blue) and lART (red) based on blood SIgA level and frequency of immunoglobulin sequences shared between the blood and the intestinal mucosa. Source data are provided as a Source Data file.

vs. 4.9% [6/101], $p < 0.0001$, respectively) (Fig. 6f). Moreover, poly-reactive B-cell antibody sequences shared between the blood and intestinal compartment were more frequently detected in lART than eART (12.1% [12/99] vs. 0.99% [1/101] $p = 0.0012$, respectively) (Fig. 6g). Among those, about two-thirds carried the IgA isotype (64% vs. 36% of IgG sequences) (Fig. 6g). Of note, data from serum SIgA levels and gut/blood-shared $V_H$ sequences alone allowed clustering separately eART and lART individuals (Fig. 6h), suggesting an increased ingress of both mucosa-derived secreted antibodies and memory B cells into the peripheral blood of lART.

## Discussion

Here, we report that IgG and IgA memory B-cell antibody repertoires in gut mucosal tissues from HIV-1-infected individuals are influenced by ART initiation timing. As previously observed[28], we show that delaying ART initiation unbalanced IgA/IgG B-cell ratio with a higher proportion of IgG-expressing B cells in lART, which may reflect mucosal inflammation as in inflammatory bowel diseases[51,52]. In humans, the vast majority of antibody-secreting cells are present in the gut mucosa and produce IgA antibodies[53]. Yet, we identified changes in immunoglobulin gene repertoires between eART and lART only for mucosal memory B cells, indicating that HIV-1 infection, and therefore treatment timing, may not profoundly modify gut plasma cell repertoires as previously reported for ART-naïve individuals leaving with HIV-1 when compared to seronegative controls[54]. These alterations in lART included higher mutation loads in IgA variable domains, despite a reduced proportion of gut IgA[+] memory B cells compared to eART. Considering the potential impairment of GALT GC in lART, this suggests clonal expansions of matured B cells following a sustained antigenic stimulation, possibly from translocating bacteria[55]. Interestingly, somatic mutations in human intestinal plasmablast-derived IgAs have been shown to be required for their microbiota cross-reactivity, which appeared often uncoupled to their polyreactivity potential[56]. Yet, mucosal memory B-cell antibodies examined here showed an overall increased poly- and commensals-reactivities in lART compared to eART. Mucosal polyreactive B cells frequently expressed $V_H4$ genes i.e., $V_H4$-34 and $V_H4$-59, which were both previously associated with bacteria- and self-reactivity in Crohn's disease and systemic lupus erythematosus[35,36,57]. In agreement, 2 out of 3 antibodies with the highest self-reactivity levels were encoded by $V_H4$-59 gene. These polyreactive mucosal antibodies reacting with taxonomically distinct intestinal bacteria strains likely contribute to preventing bacterial translocation through mechanisms such as immune exclusion[10,49,58].

Non-treated HIV-1-infected individuals are more susceptible to developing severe bacterial infection events due to their immuno-compromised status[59]. Still, early ART efficiently reduces the risk of bacterial diseases in patients with high CD4 cell counts[59]. Also, HIV-1-associated dysbiosis is characterized by an increased relative abundance of commensal Gram[-] bacteria and conversely, a decreased proportion of Gram[+] bacteria[60], and the direct implication of ART on microbiota composition in HIV-1-infected patients is documented[61].

Consistently, we found that the duration of HIV-1 infection without ART was positively correlated with circulating levels of IgA antibodies against most of the commensals tested, likely reflecting a reduced bacteremia in eART compared to lART. Conversely, serum IgG anti-body levels against *Enterococcus faecalis*, one of the most abundant Gram[+] bacteria in feces representing up to 1% of the adult intestinal microbiota[62], were higher in eART than in lART. This is congruent with anti-*E. faecalis* IgG antibodies being frequently detected in healthy individuals, but having reduced levels in HIV-1-infected individuals[63]. Overall, this contrasts with the previously-described unaltered systemic antibody responses to bacteria and microbial antigens in ART-naïve and ART-treated HIV-1-infected patients, respectively[63,64]. On the other hand, no major variations of IgG seroreactivities were evidenced between eART and lART, indicating that as opposed to inflammatory bowel diseases[65], HIV-1 infection may not be accompanied by a pro-nounced systemic IgG response to translocating gut commensal bacteria. Beside commensals-binding polyreactive mucosal antibodies, we identified a *B. subtilis* LTA-specific antibody, L2-168, which was cloned from a lART intestinal IgA[+] memory B cell. L2-168 was also found to be expressed by blood IgG[+] and IgA[+] B cells by Ig-HTS, suggesting a B-cell exit from the gut to the systemic compartment. Of note, systemic LTA levels were positively associated with HIV-1 pathogenesis and dysbiosis[60,66]. Interestingly, *B. subtilis* is a well-known probiotic bac-terium in humans that regulates gut microbiota, stimulate host immunity and reduces inflammation[67]. Whether anti-*B. subtilis* anti-bodies are commonly produced upon HIV-1 infection, and mediate pathogenic effects in the gut or instead, help blocking bacterial translocation remains to be investigated. On the other hand, although L2-168 did not cross-react with the LTA from other Gram[+] bacteria tested, we cannot exclude that it was raised in response to a more pathogenic strain than *B. subtilis*.

Polyreactive IgAs in the gut are coating most commensal bacteria, and participate in shaping and controlling the intestinal microbiota[10,68]. Mucosal IgAs, particularly dIgA or SIgA antibodies, may directly or indirectly cross the altered gastrointestinal barrier and reach the periphery in HIV-1-infected individuals as in intestinal inflammatory diseases[69,70]. Also, the direct detection of SIgA in the blood is a hallmark of mucosal antibody translocation to the periphery[69], and their serum concentration was found increased in HIV-1 seropositive individuals with IgA hyperglobulinemia[70]. Blood levels of SIgA and gut damage marker REG3α were elevated in lART compared to eART suggesting an increased translocation of mucosal antibodies into the bloodstream, presumably due to intestinal barrier leakage. Moreover, both SIgA and sCD14 levels negatively correlated with the nadir and CD4 T cell count indicating that the CD4 T-cell depletion in the blood, and possibly in the gut mucosa, has a strong impact on the ingress of mucosal antibodies and bacterial products in the peripheral blood. Whether SIgAs cross the intestinal barrier alone or in complex with opsonized bacterial components remains unclear. Considering that SIgA and sCD14 levels also positively correlated with serum IL-8 contents, which were augmented in lART, it is tempting to

speculate that SIgA immune complexes may be the predominant form of SIgA translocation in the blood. In that regard, while our assays were not sensitive enough to detect serum SIgA binding to commensals, we found a correlation between SIgA levels and total serum IgA reactivities against half of the bacteria strains tested. Chronic immune activation related to circulating microbial products is typically associated with the progression of HIV-1 infection and predicts disease outcome[30]. Yet, we did not observe any significant association between the duration of HIV-1 infection off ART and the serum IL-8 concentration. Antibodies against LPS core oligosaccharide (EndoCab) produced by T-dependent B cells may attenuate the immunostimulatory effect of LPS, and play a critical role in preventing systemic chronic inflammation[71,72], but their amounts are strongly decreased in HIV-1-infected individuals even under ART[30]. We found increased blood LPS-reactive IgA levels in lART, which were correlated with SIgA titers suggesting that a significant proportion of LPS-reactive IgAs originate from a T-independent B cell response in the gut as previously proposed[73]. In line with this, CS B cells lacking CD27 surface expression, many of them presumably emerging from T-independent B-cell responses[49], were significantly increased in the blood and in the gut of lART. We also found more elevated serum titers of anti-LPS IgA1 antibodies in lART, which could originate from both peripheral and gut lymphoid tissues. In line with this, HIV-1 infection has been recently associated to an enhanced IgA1 transcription in the gut of untreated individuals[74]. Circulating blood CD27−IgA+ memory B cells express mutated immunoglobulins/B-cell receptors and are often polyreactive[49]. They contain clones with the capacity to home to gastrointestinal tract through the expression of the homing receptor CCR9, and are therefore likely involved in the maintenance of gut homeostasis[49]. Apart from CCR9, these B cells may also express integrin α4β7 on their surface to return into inductive sites through specific interactions with the mucosal vascular addressin cell adhesion molecule 1 (MAdCAM-1)[75,76]. Using β7 as a surrogate marker for the mucosal origin of B cells, we showed that not only the level of SIgAs but also the frequency of circulating β7+CD27−IgA+ memory B cells exhibiting polyreactive binding properties were increased in lART compared to eART. Hence, our data suggest that delaying ART initiation favors the development of polyreactive intestinal memory B cells that can circulate in periphery. Human gastrointestinal tract tissues (i.e., jejunum, ileum and colon) display extensive sharing of B-cell clonal lineages[77], suggesting related B-cell repertoires across distinct intestinal locations. Moreover, B-cell memory sub-populations were shown to be similarly distributed in duodenal, jejunal and rectal tissues of rhesus macaques infected with simian-human immunodeficiency virus (SHIV)[78]. Nonetheless, we cannot ensure that the rectosigmoid-derived B-cell and plasma cell repertoires studied here are representative of those found elsewhere in the gut of ART-treated individuals leaving with HIV-1. Thus, whether HIV-1 infection also triggers the elicitation and expansion of polyreactive B-cell clones in the other anatomic sections of the intestine than the rectosigmoid segment remains to be investigated. Furthermore, we did not investigate here the intestinal microbiota composition in the donors, and how it could affect the antibody repertoire and reactivity of gut mucosal memory B cells.

We and others have shown that pre-existing poly-/commensals-reactive mucosal B cells can cross-react with Env gp41, and possibly divert the antibody response to non-neutralizing HIV-1 epitopes[23,25,79,80]. Accordingly, we found a higher proportion of circulating mucosa-derived gp140-reactive CD27−β7+ CS B cells in lART, possibly arising from GC-independent reaction[81,82]. Of note, however, comparative blood B-cell analyses between lART and eART by flow cytometry were based on a limited number of samples, which warrants additional studies on larger sized-groups to substantiate these findings. Our data also showed that a substantial fraction of commensals-binding antibodies circulating in the blood are polyreactive. Systemic antibody polyreactivity in HIV-1-infected individuals described 35-

years ago is now well documented[83,84], and was proposed to be partly due to infection-induced polyclonal B-cell activation[85], including via stimulating bacterial products such as LPS. Hence, we speculate that circulating antibodies originating from the GALT in response to translocating bacteria are an important source for the HIV-1-featured sero-polyreactivity. On the other hand, peripheral B cells responding to the translocation of bacterial products may also contribute to the production of poly- and commensals-reactive serum antibodies.

In summary, our data show that HIV-1 antiretroviral treatment timing shapes systemic and gut mucosal memory B-cell repertoires. Delaying ART initiation is associated with a heightening of poly- and commensals-reactive memory B cells in the gut mucosa, which reach the periphery and circulate in the blood where the levels of bacterial translocation, inflammation, and commensals-binding immunoglobulin markers are augmented. HIV-1-associated pathological events in the gut mucosa particularly, the translocation of commensal bacteria, trigger antimicrobial antibody and memory B-cell responses that can be attenuated by early ART. Hence, early treatment during HIV-1 primary infection prevents, in part, developing abnormal and potentially deleterious humoral responses to commensal bacteria at mucosal and systemic levels.

## Methods
### Human participants and samples
Samples were obtained as part of the research protocol called BHUANTIVIH performed in accordance with and after ethical approval from all the French legislation and regulation authorities. Biological samples were obtained from HIV-1-infected individuals under effective ART for several years at the Centre Hospitalier Regional d'Orléans (Supplementary Table 1). ART was started within 4 months after the diagnosis of primary HIV-1 infection (Fiebig stage II−V) in 38 patients (eART; median age 48 [28−79] years) or during the chronic stage of HIV-1 infection (Fiebig stage VI) in 40 patients (lART; median age 50 [23−80] years) as previously described[28]. The main clinical and immuno-virological characteristics of these donors are summarized in Supplementary Table 1. Sera were collected and analyzed for a total of 38 eART and 40 lART. Paired PBMC, sera and colon biopsies (10 rectal biopsy specimens of ~2 mm³ were collected at 10−15 cm from the anal margin) we obtained for 3 eART (E1−E3) and 3 lART (L1−L3). Intestinal biopsy (n = 8, E4−E7 and L4−L7) and PBMC (n = 3, E5, E8 and L8) samples were also obtained from additional donors (Supplementary Table 1). The clinical research protocol received approval from the *Comité Consultatif pour le Traitement de l'Information en matière de Recherche dans le domaine de la Santé* (CCTIRS) on December 12th 2013, the *Commission Nationale de l'Informatique et des Libertés* (CNIL) on August 8th 2014 and the *Comité de Protection des Personnes de Tours* (CPP Région Centre-Ouest 1) on December 17th 2014. All donors gave written consent to participate in this study, and data were collected under pseudo-anonymized conditions using subject coding. Ethical issues have been monitored by the Ethics Board for European contracts, an ad hoc independent Ethics Committee in charge of reviewing periodically sensitive ethical issues in EU funded research when requested by the EU.

### Cell isolation
Intraepithelial lymphocytes (IEL) were isolated by two rounds of vigorous shaking in DMEM-Glutamax (#61965-059, Gibco, Thermo Fisher Scientific) supplemented with 1% Fetal Bovine Serum (FBS) (#10309433, Gibco), 1 mM EGTA and 1.5 mM MgCl₂. Lamina propria lymphocytes (LPL) were isolated by two rounds of tissue digestion in medium containing collagenase II 100U/ml (#C1764-50MG, Sigma-Aldrich), followed by mechanical disruption with a syringe equipped with a 16-gauge blunt-end needle. IEL and LPL were pooled and washed in DMEM−Glutamax (Gibco) supplemented with 10% FBS (Gibco), and 1% penicillin/streptomycin (#11548876, Gibco). PBMC were isolated

from donors' blood using Ficoll Plaque Plus (#11743219, GE Healthcare).

### Serum IgG and IgA purification

Sera were heat-inactivated at 56 °C for 1 h. Human IgG and IgA antibodies were purified from donors' sera by affinity chromatography using Protein G Sepharose® 4 Fast Flow (#90100093, GE Healthcare) and peptide M-coupled agarose beads (#gel-pdm-5, Invivogen), respectively. Purified serum antibodies were dialyzed against PBS using Slide-A-Lyzer® Cassettes (#66380, 10 K MWCO, Thermo Fisher Scientific). Protein concentrations were determined using a NanoDrop 2000 instrument (Thermo Fisher Scientific).

### Antigens and control antibodies

Clade B YU-2 trimeric foldon-type gp140 (gp140-F) and gp120 proteins[86] were produced by transient transfection of exponentially growing Freestyle™ 293-F suspension cells (#R79007, Thermo Fisher Scientific) using polyethylenimine (PEI, #23966-2, Polysciences)-precipitation method, purified by high-performance chromatography using the Ni Sepharose® Excel Resin according to manufacturer's instructions (#17371202, GE Healthcare), and controlled for purity by SDS-PAGE and NativePAGE gel staining as previously described[87,88]. The trimeric state of purified YU-2 gp140 was further confirmed by size exclusion FPLC-chromatography using an AKTA pure FPLC instrument (GE Healthcare) with a Superdex® 200 increase 10/300 GL column (GE Healthcare). AviTaged YU-2 gp140-F trimers used for B-cell FACS were biotinylated using BirA biotin-protein ligase bulk reaction kit (Avidity, LLC). Purified clade B MN gp41 protein (#12027) and HXB2 p24 (#13126) were provided by the NIH AIDS Reagent Program. The *E. coli* RNA polymerase α subunit (RNApol α) was produced and purified[25]. Insulin (#I9278-5ML, Sigma-Aldrich) and RNApol α were coupled to DyLight 650 (#62265) and DyLight 405 (#46400), respectively, using the DyLight® Amine-Reactive Dyes kit (Thermo Fisher scientific). Polyreactive and non-polyreactive antibody ED38[89] and mGO53[90], were produced as described below.

### Single B-cell flow cytometry phenotyping, sorting and expression-cloning of antibodies

Isolated mucosal cells were first stained using LIVE/DEAD aqua fixable dead cell stain kit (Molecular Probes, Thermo Fisher Scientific) to exclude dead cells. Cells were then incubated for 30 min at 4 °C with biotinylated YU2 gp140-F trimers[91,92], washed once with 1% FBS-PBS (FACS buffer), and incubated for 30 min at 4 °C with a cocktail of mouse anti-human antibodies: CD19 Alexa 700 (HIB19, #557921, BD Biosciences, 1:100), CD38 APC (HIT2, #560980, BD Biosciences, 1:100), CD21 BV421 (B-ly4, #562966, BD Biosciences, 1:100), CD27 PE-CF594 (M-T271, #562297, BD Biosciences, 1:100), IgM BV605 (G20-127, #562977, BD Biosciences, 1:100), IgG BV786 (G18-145, #564230, BD Biosciences, 1:100), IgA FITC (IS11-8E10, #130.114.001 Miltenyi Biotec, 1:100), Integrin β7 BUV395 (FIB504, #744014, BD Biosciences, 1:100) and streptavidin R-PE conjugate (#SA10041, Invitrogen, Thermo Fisher Scientific, 1:1000). Single mucosal CD19⁺IgA⁺ and CD19⁺IgG⁺ class-switch memory B cells (CD27⁺CD38⁻) and antibody-secreting cells (CD27⁺CD38⁺) from e-ART (*n* = 3) and l-ART (*n* = 3) donors were single-cell sorted into 96-well PCR plates using a FACS Aria II sorter (Becton Dickinson) as previously described[88]. Single-cell cDNA synthesis using SuperScript IV reverse transcriptase (Thermo Fisher Scientific) followed by nested-PCR amplifications of IgH, Igκ and Igλ genes, and sequences analyses for Ig gene features were performed as previously described[88]. Purified digested PCR products were cloned into human Igγ₁-, Igκ- or Igλ-expressing vectors (GenBank# LT615368.1, LT615369.1 and LT615370.1, respectively) as previously described[88]. Recombinant IgG1 antibodies were produced by transient co-transfection of Freestyle™ 293-F suspension cells (Thermo Fisher Scientific) using PEI-precipitation method as previously described[25]. Monoclonal

antibodies were purified by batch/gravity-flow affinity chromatography using protein G sepharose 4 fast flow beads (#90100093, GE Healthcare) and dialyzed against PBS.

### Flow cytometry B-cell binding assay

PBMC were first stained using LIVE/DEAD aqua fixable dead cell stain kit (#L34957, Molecular Probes, Thermo Fisher Scientific) to exclude dead cells. Cells were then incubated for 30 min at 4 °C with biotinylated HIV-1 gp140 trimer, DyLight 650-coupled Insulin (10 µg/ml), DyLight 405-coupled RNA polymerase α (10 µg/ml), and Alexa Fluor™ 594-coupled LPS (10 µg/ml, #L23353, Thermo Fisher scientific), washed once with 1% FBS-PBS (FACS buffer), and incubated for 30 min at 4 °C with a cocktail of mouse anti-human antibodies: CD19 Alexa 700 (HIB19, #557921, BD Biosciences, 1:100), CD21 BV421 (B-ly4, #562966, BD Biosciences, 1:100), CD27 BV711 (M-T271, #564893, BD Biosciences, 1:100), IgG BV605 (G18-145, #563246, BD Biosciences, 1:100), IgA FITC (IS11-8E10, #130.114.001, Miltenyi Biotec, 1:100), Integrin β7 BUV395 (FIB504, #744014, BD Biosciences, 1:100) and streptavidin R-PE conjugate (#SA10041, Invitrogen, Thermo Fisher Scientific, 1:1000). Cells were then washed and resuspended in FACS buffer. Following a lymphocyte and single-cell gating, live cells were gated on CD19⁺ B cells. FACS analyses were performed using a BD LSR Fortessa instrument (Becton Dickinson) and FlowJo software (v10.3, FlowJo LLC). Polyreactive cells were defined as B cells binding to fluorescently labeled Insulin among those reactive to both RNA polymerase α and LPS (polyR; Insulin⁺ RNA pol α⁺ LPS⁺). To ensure binding specificity to gp140, gp140⁺ cells were gated on non-polyR B cells (Insulin⁻ RNA pol α⁻ LPS⁻).

### High-throughput immunoglobulin sequencing

Total mRNAs were extracted from PBMC and intestinal cells using Nucleospin RNA kit and Nucleospin RNA XS kit (Macherey Nagel), respectively. cDNAs were generated from 0.5–1 µg mRNAs using random hexamers (pd(N)6, Roche) and SuperScript IV reverse transcriptase (Thermo Fisher Scientific) following the manufacturer procedures. Variable domain IgG and IgA DNA fragments were first amplified from 5 µl of cDNA using 2 U of Platinum™ Taq DNA polymerase (Thermo Fisher Scientific) according to manufacturer's instructions, and 5'L-VH mix and 3'CγCH1 or 3'CαCH1 primers[93], with the following PCR cycles: 2 min at 94 °C, 50 x [94 °C for 30 s–58 °C for 30 s–72 °C for 1 min], and 72 °C for 5 min. PCR products were gel-purified using the NucleoSpin® Clean-up kit (Macherey Nagel), and subjected to an additional PCR-amplification (5 µl of template) following the aforementioned conditions but using 5'L-VH mix and 3'IgGint or 3'CαCH1-2 primers[93]. Amplified products were gel-purified using the NucleoSpin® Clean-up kit (Macherey Nagel). DNA quantification, DNA library preparation and sequencing were performed at the Biomics platform (Institut Pasteur) using Illumina technology. Briefly, DNA libraries were quantified and quality-checked using 5200 Fragment Analyzer System (Agilent Technologies), and then prepared using NEXTflex™ PCR-Free DNA Sequencing Kit (Bioo Scientific) according to manufacturer's instructions (no fragmentation step). DNA fragments were loaded on the flow cell [20% Phix at 12.5 pM and 80% of library pool at 6 pM (v/v)], and sequenced using MiSeq® Reagent Kit v3 (Illumina) and Miseq instrument, which generated 2x300-bp paired-end reads.

### Bioinformatics pipeline and analyses

Forward and reverse reads with 10 overlapping nucleotides minimum were merged using PEAR software[94] with the following parameters: minimum length $n = 300$, quality threshold $q = 20$, and minimum length after trimming low-quality parts $t = 300$. Assignment of immunoglobulin germline genes was performed using IgBlast[95] (https://www.ncbi.nlm.nih.gov/igblast/). Isotype definition was made using a BLAST procedure. A database was built using an inhouse program and

sequences presenting low-quality sequence features (Stop codon, V-gene $E$-value > $10^{-3}$, undefined $CDR_H3$ ending motif WGXG, and subject alignment start >9) were removed. We then used CD-HIT[96] (http://weizhongli-lab.org/cd-hit/) to cluster similar V-REGION sequences sharing a 99% identity to remove redundancy and only the representative sequence was used. The clonal families were established by grouping sequences that: (1) had the same V and J genes, (2) shared the same $CDR_H3$ length, (3) and had cutoff distance (≥0.8) amongst $CDR_H3$ amino acid sequences. The diversity visualization was built following the network approach developed by Bashord-Rodgers et al.[97]. Each vertex represents a unique sequence with its size proportional to the number of identical read sequences to the vertex. Edges connecting two vertices were drawn only if they differed by a single non-indel nucleotide. The networks were computed using the igraph library implemented in R (https://igraph.org/r/). The network graph provides an intuitive picture of the clone size distribution of the BCR population.

## ELISAs

Antibody binding to HIV-1 Env proteins were measured by ELISA as previously described[91,98]. Briefly, high-binding 96-well ELISA plates (Costar, Corning) were coated overnight with 125 ng/well of HIV-1 Env proteins. After washings, plates were blocked 2 h with 2% BSA, 1 mM EDTA, 0.05% Tween 20-PBS (Blocking buffer), washed, and incubated with serially diluted purified serum IgA/IgG and recombinant monoclonal antibodies in PBS. Polyreactivity ELISA was performed as previously described[25]. Briefly, high-binding 96-well ELISA plates were coated overnight with 500 ng/well of purified double stranded (ds)-DNA (#D1501-100MG, Sigma-Aldrich), KLH (#H8283-50MG, Sigma-Aldrich), LPS (#L2637-5MG, Sigma-Aldrich), 250 ng/well of insulin (#I9278-5ML, Sigma-Aldrich), and 125 ng/well of HIV-1 proteins in PBS. After blocking and washing steps, recombinant monoclonal IgG and purified serum antibodies were tested at 4 µg/ml and 50 µg/ml, respectively, and 3 consecutive 1:4 dilutions in PBS. Control antibodies, mGO53 (negative), and ED38 (high positive) were included in each experiment. Binding of human monoclonal IgG antibodies to HEp-2 cell-expressing autoantigens were analyzed in triplicate at 4 µg/ml by ELISA (AESKULISA® ANA-HEp-2, Aesku.Diagnostics). ELISA testing to bacteria strains was performed as follows: *E. coli* (CIP 54.8T), *E. faecalis* (CIP 103015T), *E. cloacae* (CIP 60.85T), *M. morganii* (CIP 231T), *B subtilis* (CIP 52.65T), *C. difficile* (CIP 104282T), *P. corporis* (CIP 105107T), *B. cereus* (CIP 66.24T), *B. licheniformis* (CIP 52.71T) and *B. cytotoxicus* (CIP 110041T) were obtained from the biological resource center of Institut Pasteur (CRBIP). Bacteria were grown in LB medium from a single colony up to a culture $OD_{600\,nm}$ of 1, fixed with 0.2% paraformaldehyde (Sigma) for 20 min, washed, and coated on Poly-L-Lysine-treated high-binding 96-well ELISA plates as previously described[12]. Recombinant monoclonal IgG1 and purified serum antibodies were tested in duplicate at 4 µg/ml as described above. The reactivity of the recombinant monoclonal antibody L2-168 against *B. subtilis* peptidoglycan (#69554-10MG-F, Sigma-Aldrich) and flagellin (#tlrl-bsfla, Invivogen), and against other *Bacillaceae* was also evaluated in triplicate at 50 µg/ml and 20 µg/ml, respectively, and 7 consecutive 1:3 dilutions in PBS as described above. To measure serum dimeric (dIgA) and secretory IgA (sIgA) antibody levels, plates were coated overnight with 125 ng/well of mouse anti-human J chain (Mc19-9, #MCA693, BIO-RAD) and anti-human secretory component (GA-1, #I6635, Sigma-Aldrich) antibodies, respectively. ELISA were then performed with purified serum IgA antibodies tested at 100 µg/ml and 3 consecutive 1:3 dilutions in PBS in duplicate as described above. After washings, the plates were revealed by incubation for 1 h with goat HRP-conjugated anti-human IgG (#109-035-098) or anti-human IgA antibodies (#109-035-011) (Jackson ImmunoResearch, 0.8 µg/ml final) and by adding 100 µl of HRP chromogenic substrate (ABTS solution, #11001-AAT, Euromedex) after washing steps. Optical densities were measured at 405 nm ($OD_{405nm}$),

and background values given by incubation of PBS alone in coated wells were subtracted. Experiments were performed using Hydro-Speed™ microplate washer and Sunrise™ microplate absorbance reader (Tecan Männedorf). Serum Reg3α (#DY5940-05), soluble CD14 (sCD14, #DC140) and IL-8 (#D8000C) levels were quantified using DuoSet ELISA kits (R&D systems) according to the manufacturer' instructions.

## HIV-1 virus production and in vitro neutralization assay

HIV-1 Env plasmids (Bal.26, YU2, PVO.4 and WITO4160.33) were obtained from the NIH AIDS Reagent Program. Pseudoviruses were prepared by co-transfection of HEK-293T cells (CRL-11268™, ATCC) with pSG3ΔEnv vector using FUGENE-6 transfection reagent (Promega) as previously described[99,100]. Pseudovirus-containing culture supernatants were harvested 48 h post-transfection, and 50% tissue culture infectious dose (TCID50) of each preparation was determined using TZM-bl cells (#8129, NIH AIDS Reagent Program) and a Wallac 142 VICTOR2 microplate reader (Perkin Elmer) as previously described[99,100]. Neutralization % of cell-free HIV-1 were calculated as follows: [100 − ((RLU IgG sample at 250 µg/ml − RLU cells alone) / (RLU cells with virus − RLU cells alone) × 100)]. Neutralization experiments were performed in triplicate and included mGO53 (negative) and 10-1074 (positive)[98] controls.

## Indirect immunofluorescence assays

Recombinant IgG antibodies (150 µg/ml final concentration) were assayed by indirect immunofluorescence assay (IFA) using the ANA HEp-2 AeskuSlides® kit (Aesku.Diagnostics) following the manufacturer's instructions. mGO53, ED38 and kit's control antibodies were included in each experiment. Binding of human IgG antibody L2-168 to *B. subtilis* was tested by IFA. Briefly, fixed bacteria were 1:10-diluted in PBS and incubated overnight on Poly-L-Lysine-treated Superfrost Plus Slides (Fischer Thermo Scientific). Slides were then washed twice with PBS and incubated with L2-168 at 10 µg/ml for 1 h at room temperature. After washing steps, FITC-conjugated anti-human IgG antibodies from ANA HEp-2 AeskuSlides® kit was used as tracer for revealing bacteria binding. IFA slides were examined using the fluorescence microscope Axio Imager 2 (Zeiss), and pictures were taken at magnification ×100 with 5000 ms-acquisition using ZEN imaging software (Zen 2.0 blue version, Zeiss) at the Imagopole platform (Institut Pasteur).

## Dot-blotting

Lipoteichoic acids (LTA; 0.3, 3, 30 and 300 ng) from *B. subtilis* (#L3265-5MG, Sigma-Aldrich), *S. aureus* (#L2515-5MG, Sigma-Aldrich) and *S. pyogenes* (#L3140-5MG, Sigma-Aldrich) were immobilized on nitrocellulose membranes for 2 h at room temperature. Membranes were then saturated for 2 h in PBS-0.05% Tween 20 (PBST)-5% dry milk and incubated for 2 h with L2-168 antibody at 10 µg/ml in PBST-5% dry milk. After washings with PBST, membranes were incubated for 1 h with 1/25,000-diluted Alexa Fluor 680-conjugated donkey anti-human IgG (#709-625-149, Jackson ImmunoReseach) in PBST-5% dry milk. Finally, membranes were washed, and examined with the Odyssey Infrared Imaging system (LI-COR Biosciences).

## Protein microarray binding analyses

All experiments were performed at 4 °C using ProtoArray Human Protein Microarrays (Thermo Fisher Scientific). Microarrays were blocked for 1 h in blocking solution (Thermo Fisher), washed and incubated for 1 h 30 min with IgG antibodies at 2.5 µg/ml as previously described[25]. After washings, arrays were incubated for 1 h 30 min with AF647-conjugated goat anti-human IgG antibodies (at 1 µg/ml in PBS; # A-21445, Thermo Fisher Scientific), and revealed using GenePix 4000B microarray scanner (Molecular Devices) and GenePix Pro 6.0 software (Molecular Devices) as previously described[25]. Fluorescence intensities were quantified using Spotxel® software (SICASYS Software GmbH),

and mean fluorescence intensity (MFI) signals for each antibody (from duplicate protein spots) was plotted against the reference antibody mGO53 (non-polyreactive isotype control) using GraphPad Prism software (v8.1.2, GraphPad Prism Inc.). For each antibody, Z-scores were calculated using ProtoArray® Prospector software (v5.2.3, Thermo Fisher Scientific), and deviation (σ) to the diagonal, and polyreactivity index (PI) values were calculated as previously described[25]. Antibodies were defined as polyreactive when PI > 0.21.

## Statistics

The numbers of $V_H$, Vκ and Vλ mutations were compared across groups of antibodies using unpaired two-tailed Student's $t$ test with Welch's correction. Bivariate correlations were assayed using two-tailed Pearson correlation test. Antibody reactivities (sera and monoclonals) and levels of serum markers were compared between groups using two-tailed Student's $t$ test with Welch's correction. Frequencies of blood B-cell sub-populations were compared between groups using two-tailed Mann–Whitney test. Cumulative distributions of antimicrobial reactivities between gut-derived monoclonal and serum IgG antibodies were compared using the Kolmogorov–Smirnov unpaired test. Statistical and analyses were performed using GraphPad Prism software (v8.2, GraphPad Prism Inc.). Volcano plots comparing gene features ($n$ = 206 parameters) of intestinal B cells was also performed using GraphPad Prism software (v8.2, GraphPad Prism Inc.). The $y$ axis indicates the statistics expressed as $-\log_{10}$ ($p$ values) and the $x$ axis represents the differences between the group means for each parameter. Principal component analysis (PCA) was performed using the Qlucore Omics Explorer bioinformatics software. The Barnes-Hut implementation of $t$-distributed stochastic neighbor embedding (t-SNE) was computed using FlowJo software (v10.3, FlowJo LLC, Ashland, OR) with 2000 iterations and a perplexity parameter of 200. Colors represent density of surface expression markers or cell-populations varying from low (blue) to high (red). The correlation plot between anti-p24 and -g140 antibody reactivities (Fig. 2a) was generated using FlowJo software (v10.3; FlowJo LLC) after transforming for all final sequences in a given library, (x, y) coordinates-containing text file into a FCS format file using DISCit software[101]. The correlation matrix of clinico-virological and serological values was generated with R software (v4.1.3) using rcorr() function from Hmisc package (v4.7, https://CRAN.R-project.org/package=Hmisc). The correlogram was generated using the corrplot() function from corrplot package (v0.92, https://CRAN.R-project.org/package=corrplot). Cells are colored according to the correlation values between two parameters (blue colors: positive correlations, white: no correlation and pink: negative correlations). Principal component analysis (PCA) was performed with R software (v4.1.3) using the PCA() function from FactoMineR package (v2.4, https://CRAN.R-project.org/package=FactoMineR). PCA plot [fviz_pca_biplot()], was generated using the factoextra package (v1.0.7, https://CRAN.R-project.org/package=factoextra).

## Reporting summary

Further information on research design is available in the Nature Portfolio Reporting Summary linked to this article.

## Data availability

All statistical values for the multiparametric analyses performed and presented in the study are provided in the Source Data file. Source data are provided with this paper.

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

## Acknowledgements
We are grateful to all participants who consented to be part of this study. We thank Sandrine Schmutz and Sophie Novault for their help with single-cell sorting (CB-UTechS, Institut Pasteur), Sylviane Hamon for providing commensal bacteria (CRBIP, Institut Pasteur), Laurence Ma, Christiane Bouchier and Cedric Fund for performing the NGS experiments (Biomics, Institut Pasteur). We also thank the NIH AIDS Reagent Program (division of AIDS, NIAID, NIH) for contributing reagents and the Agence National de Recherche sur le SIDA et les hépatites virales (ANRS) for an equipment grant. H.M. received core grants from the G5 Institut Pasteur Program, the *Milieu Intérieur* Program (ANR-10-LABX-69-01) and INSERM. This work was funded by the European Research Council (ERC)—Seventh Framework Program (ERC-2013-StG 337146). S.H. and H.M. received a grant from SIDACTION (15-1-AEQ-05-02) for this study. J.D.D. received a grant from the European Research Council (ERC) Horizon 2020-work Program (ERC-2015-StG-678905). C.P. was supported by a 1-year postdoctoral fellowship from the ANRS.

## Author contributions
C.P., S.H., L.H. and H.M. conceived the study framework. H.M. supervised the study. C.P. and H.M. designed the experiments. C.P., L.M.M.-A., A.K., J.D.D. and H.M. performed and analyzed the experiments. P.R. and T.H. performed bioinformatic analyses. D.C., T.P., L.L., S.H. and L.H. provided key biological reagents and human samples. C.P. and H.M. wrote the manuscript with contributions from all the authors.

## Competing interests
The authors declare no competing interests.
