## [Peer Review File · Nature Communications]

HIV-1 Treatment Timing Shapes the Human Intestinal Memory B-Cell Repertoire to Commensal BacteriaREVIEWER COMMENTS

Reviewer #1 (Remarks to the Author):

Mouquet and colleagues investigate how the timing of ART initiation influences intestinal memory B cell and plasma cell repertoires obtained by rectosigmoid biopsies. They find increases in polyreactive and commensal specific IgA and IgG memory B cells in individuals that started ART during chronic infection. Their findings are correlated with serologic activity and systemic measures of inflammation and intestinal permeability.

The manuscript is impressive in the detailed analysis of the specificity of the antibodies being produced in the gut in early vs. chronic therapy. 200 randomly selected memory antibodies were produced and tested for specificity and only 2 bound to the HIV-1 spike or p24. Polyreactivity was tested with a variety of different antigens revealing a significant increase in the chronics and an overall high level of polyreactivity with an increase prevalence of VH4-34 and 4-39.

Overall this is an outstanding collection of data relevant to understanding the effects of starting ART early vs. late. It argues strongly that ART should be started early in order to minimize the destructive effects of the virus on gut immune physiology and more general associated inflammation.

Reviewer #2 (Remarks to the Author):

The authors of this manuscript present evidence that delayed initiation of antiretroviral therapy (ART) in people with HIV infection (PWH) has a detrimental effect on gut mucosal B cell/antibody responses, which contributes to dysfunction of circulating B cells/antibodies and immune activation. By examining monoclonal antibodies derived from class switched memory B cells isolated from recto-sigmoid mucosa, they observed that PWH who commenced ART later exhibited differences in the abundance of V and J genes, increased frequencies of some CDR3 sequences and VH and Vk somatic mutations, particularly in IgA+ cells, and an accumulation of polyreactive and anti-gut bacteria B-cell clones in the gut. Other investigations demonstrated that PWH receiving late ART also exhibited augmented IgA antibody responses against gut bacteria and LPS, antibody polyreactivity that was highest amongst IgA+ B cells, and antibody self-reactivity, again mainly from IgA+ B cells, though not with IFA on Hep2 cells. Furthermore, polyreactive B cells were enriched amongst IgA+/CD27-/β7+ B cells in late ART patients compared with early ART patients and there were much higher proportion of B cells with mucosal and polyreactive antibody sequences in blood of PWH receiving late ART than those receiving early ART. Finally, serum levels of dimeric and secretory IgA and markers of systemic inflammation and bacterial translocation were higher and related to polyreactive blood memory B-cell frequencies in PWH receiving late ART.

The study reported in the manuscript is very comprehensive and has been undertaken with a high degree of scientific rigour. The data are very relevant to an increased understanding of HIV-associated immune activation and aberrant neutralising antibody response against HIV-1. The quality of the manuscript and diagrams is very high. There are, however, several issues that the authors should consider:

1. The study findings contribute to the body of knowledge on mechanisms of HIV-1-associated immune activation but other possible outcomes of increased B cell/antibody polyreactivity in untreated HIV-1 infection could be discussed in greater depth. This is a very interesting aspect of the findings of this study. For example, are the authors suggesting in lines 407-409 that induction of T-independent mucosal antibody responses against commensal bacteria that cross-react with HIV-1 gp41, thereby diverting anti-HIV-1-Env antibody responses away from neutralising epitopes, is an immune evasion strategy of HIV-1? Do ineffective anti-HIV-1 neutralising antibody responses reflect, at least in part, cross-reactive B cell/antibody responses arising in gut-associated lymphoid tissue, which is a major site of HIV-1 replication? In addition, are these anti-gp41 antibodies related to those that exhibit polyreactivity with phospholipids and other autoantigens (see, for example, Dennison SM et al. J Virol. 2011; 85:1340-7).

2. The authors have highlighted the importance of IgA and IgG antibody responses in gut mucosa being either T-independent or T-dependent but have not examined this in any great detail in their study. For example, are higher serum levels of IgG anti-Enterococcus faecalis antibodies in PWH receiving early ART related to higher CD4+ T cell counts, while serum IgA anti-LPS antibodies are unrelated to CD4+ T cell counts.

3. The suggestion “that a significant proportion of LPS-reactive IgAs originate from a T-independent B cell response in the gut.” (lines 388-90) is supported by previously published data (see - Lim A et al. AIDS. 2011; 25:1379-83).

4. How representative are rectosigmoid MBCs and plasma cells of similar cells in other parts of the gut? Do the authors have data of their own showing that they are representative or can they refer to data of other investigators? If not, it should be indicated that this is a possible limitation of this study.

5. Minor issues:

a) In line 46, the authors refer to IgD antibodies. What evidence is there of IgD antibodies in gut mucosa?

b) In line 347, insert word ‘of’ between ‘one’ and ‘the’.

Reviewer #3 (Remarks to the Author):

Planchais and colleagues investigated the effect of beginning ART during early vs chronic HIV infection on mucosal B cells and related population in peripheral blood. The authors make several important observations but a few are not sustained by the data presented. In addition, several observations are made without interpretation.

Specific points:

1. Table S1 is confusing: the chronic (L) group is defined by starting ART in chronic phase of infection as referenced in another paper but several in the L group seem to belong to the E group based on when they started ART, some at the time of infection. Are the authors conflating time of diagnosis with time of infection?

2. The authors state on line 114 that they carried out deep sequencing on matched colorectal and peripheral blood samples, indicating analysis of 7 in each group yet in the first part of the study, line 88, they state a total of 6 colorectal biopsies were performed (3 in each

group). Which is it or were only 3 analyzed by single-cell and extended to 4 others for bulk analyses? This should be clarified. More importantly, L4 and L7 seem to have started ARVs shortly after infection which would categorize them as early. Again, is this a difference between time of diagnosis vs time of infection?

3. The mucosal IgG mutational load for VH does not appear to agree between single cell (Fig. 1E) and bulk (Fig. 1F) analyses yet authors state these were similar, line 117. The differences were modest yet highly significant for the latter but not the former. Does this indicate that single-cell analyses were not representative? This needs to be addressed.

4. Do authors have an explanation why the 2 HIV-specific antibodies would bind both p24 and gp140. Did this extend to gp120 and/or gp41? Further, how did the evidence of polyreactivity in Figures 2B and S3 break down by group and for intestinal mAbs reconstituted from people who are not HIV-infected?

5. Is it common to see such differences as shown in Fig 2D and 2F between Hep-2 ELISA and IFA? What might be the explanation?

6. Line 198-206 discusses one mAb, L2-168 that has a non-polyreactive specificity. Why did the authors feel this was of importance? This is already a long paper and this result does not appear to add any value to the overall message. The authors should consider removing this part.

7. Is it common for LPS reactivity to be driven by IgA1 over IgA2 when latter is the predominant source in the lower gastrointestinal tract?

8. From line 227: if striking, why not add Fig. S4E to main figure in lieu of Fig. 2D-F?

9. Lines 237-248: Regarding elevated dIgA and SIgA, this may simply reflect overall increases in IgA that is associated with chronic HIV. Do these specific differences hold if corrected for differences in total IgA?

10. The most important point: the authors do not make a clear case for gut-related polyreactive blood CD27- IgA+ memory B cells in IART as stated in subtitle line 273 and Abstract line 30 and Discussion line 398. It is very unfortunate that the authors did not test more subjects to address questions posed starting line 275. It is understandable that colorectal biopsies are not routine procedures, but blood collection and phenotypic analyses are. The data shown in Figure 5B did not reach statistical significance, there was no increase in circulating IgA B cells that express the mucosal markers CD27-B7+ and Figure 5C showing a higher proportion of gp140+ B cells among CD27-B7+ in IART is based on very few events and must be confirmed with more sampling. The authors refer to polyreactive clones, beginning line 294, among circulating B cells based on binding to three proteins. It is not clear how the triple binding was determined based on the flow plots shown in Fig. S5A and it cannot be assumed that these proteins are binding to the BCR unless mAbs are cloned. The direct evidence of connection between peripheral and mucosal B cells comes from the sequences in Ig-HTS libraries from PBMCs that matched with mucosal mAb sequences. However, the polyreactive sequences were not broken down by IgA vs IgG which questions how much of the enrichment in IART is driven by IgA?

11. Overall, the authors show elements of increased polyreactivity of IgA mucosal origin and connection between blood and intestinal sequences but there is no clean link to a population in the peripheral blood. The authors need to address this lack of connection.

12. In the Discussion, line 322-324, the authors do not account for differences between their cohort and the one in reference 53: the latter was conducted on PLWH who were viremic at time of study. This statement should be modified accordingly.

Minor points

1. IgG on surface of ASC can be low and difficult to evaluate without permeabilization. Were

all ASC accounted for by an Ig isotype? If not, then it would suggest that those missing may be IgG and undercounted.

2. For readability, add LPS to Fig 3C.

3. Lines 139-141: it cannot be that surprising to have a very low frequency of HIV-specificity among the total memory B-cell repertoire for PLWH on ART. This low frequency does not necessarily suggest polyreactivity. The authors should not make such an assumption.

We would like to thank the reviewers for taking the time for a critical reading of our manuscript, and for their thoughtful comments. Please find below our detailed point-by-point response to the three referees.

Reviewer #1 (Remarks to the Author):

Mouquet and colleagues investigate how the timing of ART initiation influences intestinal memory B cell and plasma cell repertoires obtained by rectosigmoid biopsies. They find increases in polyreactive and commensal specific IgA and IgG memory B cells in individuals that started ART during chronic infection. Their findings are correlated with serologic activity and systemic measures of inflammation and intestinal permeability.

The manuscript is impressive in the detailed analysis of the specificity of the antibodies being produced in the gut in early vs. chronic therapy. 200 randomly selected memory antibodies were produced and tested for specificity and only 2 bound to the HIV-1 spike or p24. Polyreactivity was tested with a variety of different antigens revealing a significant increase in the chronics and an overall high level of polyreactivity with an increase prevalence of VH4-34 and 4-39.

Overall this is an outstanding collection of data relevant to understanding the effects of starting ART early vs. late. It argues strongly that ART should be started early in order to minimize the destructive effects of the virus on gut immune physiology and more general associated inflammation.

We are thankful to the referee for her/his very positive global comments.

Reviewer #2 (Remarks to the Author):

The authors of this manuscript present evidence that delayed initiation of antiretroviral therapy (ART) in people with HIV infection (PWH) has a detrimental effect on gut mucosal B cell/antibody responses, which contributes to dysfunction of circulating B cells/antibodies and immune activation. By examining monoclonal antibodies derived from class switched memory B cells isolated from recto-sigmoid mucosa, they observed that PWH who commenced ART later exhibited differences in the abundance of V and J genes, increased frequencies of some CDR3 sequences and VH and Vk somatic mutations, particularly in IgA+ cells, and an accumulation of polyreactive and anti-gut bacteria B-cell clones in the gut. Other investigations demonstrated that PWH receiving late ART also exhibited augmented IgA antibody responses against gut bacteria and LPS, antibody polyreactivity that was highest amongst IgA+ B cells, and antibody self-reactivity, again mainly from IgA+ B cells, though not with IFA on Hep2 cells. Furthermore, polyreactive B cells were enriched amongst IgA+/CD27-/β7+ B cells in late ART patients compared with early ART patients and there were much higher proportion of B cells with mucosal and polyreactive antibody sequences in blood of PWH receiving late ART than those receiving early ART. Finally, serum levels of dimeric and secretory IgA and markers of systemic inflammation and bacterial translocation were higher and related to polyreactive blood memory B-cell frequencies in PWH receiving late ART.

The study reported in the manuscript is very comprehensive and has been undertaken with a high degree of scientific rigour. The data are very relevant to an increased understanding of HIV-associated immune activation and aberrant neutralising antibody response against HIV-1. The quality of the manuscript and diagrams is very high. There are, however, several issues that the authors should consider:

We thank the reviewer for her/his very positive feedback on the manuscript.

Questions/comments:

1. The study findings contribute to the body of knowledge on mechanisms of HIV-1-associated immune activation but other possible outcomes of increased B cell/antibody polyreactivity in untreated HIV-1 infection could be discussed in greater depth. This is a very interesting aspect of the findings of this study. For example, are the authors suggesting in lines 407-409 that induction of T-independent mucosal antibody responses against commensal bacteria that cross-react with HIV-1 gp41, thereby diverting anti-HIV-1-Env antibody responses away from neutralising epitopes, is an immune evasion strategy of HIV-1? Do ineffective anti-HIV-1 neutralising antibody responses reflect, at least in part, cross-reactive B cell/antibody

responses arising in gut-associated lymphoid tissue, which is a major site of HIV-1 replication? In addition, are these anti-gp41 antibodies related to those that exhibit polyreactivity with phospholipids and other autoantigens (see, for example, Dennison SM et al. J Virol. 2011; 85:1340-7).

This is a very interesting point of particular interest for us, but completely unresolved so far. Unfortunately, indeed, only few studies (principally from the B.F. Haynes group and ours) have reported the existence of gut polyreactive B-cell cross-reacting with gp41, which could be responsible for a potential bias of the humoral response and a diversion to non-neutralizing HIV-1 epitopes. However, absolute demonstrations of this phenomenon have not been fully provided yet. Thus, one cannot claim that this is an HIV-1 evasion strategy without further evidence, but it is certainly a deleterious effect explained by the nature of both, the gut B-cells – many of which being polyreactive, and the HIV-1 Env particularly gp41 being very good ligands for cross-/poly-reactive B cells and antibodies. But yes, we and the Haynes's team posit, based on our respective studies, that there is a link between ineffective functional humoral responses to HIV-1 in the GI tract and the highly polyreactive B cells populating these tissues. Yes, we consider that the intestinal polyreactive B cells we described here as being more frequently found in IART could be the precursors for the B cells from which the antibodies described in Dennison et al (2011) arose. We have added this reference on page 17, line 410.

2. The authors have highlighted the importance of IgA and IgG antibody responses in gut mucosa being either T-independent or T-dependent but have not examined this in any great detail in their study. For example, are higher serum levels of IgG anti-Enterococcus faecalis antibodies in PWH receiving early ART related to higher CD4+ T cell counts, while serum IgA anti-LPS antibodies are unrelated to CD4+ T cell counts.

We thank the reviewer for her/his suggestions. In accordance, we performed correlation analyses between the serum levels of anti-Enterococcus faecalis IgG and anti-LPS IgA antibodies versus CD4+ T cell counts (see figure on the right). Despite a trend for a positive correlation between anti-E. faecalis IgG antibodies and CD4+ T cell counts in eART and for a negative correlation between serum anti-LPS IgA antibodies and CD4+ T cell counts in IART, we did not observe any significant differences. In consequence, we chose not to include those data in the manuscript.

3. The suggestion “that a significant proportion of LPS-reactive IgAs originate from a T-independent B cell response in the gut.” (lines 388-90) is supported by previously published data (see - Lim A et al. AIDS. 2011; 25:1379-83).

We thank the reviewer for pointing out this work, which is now cited as a reference for this suggested possibility.

4. How representative are rectosigmoid MBCs and plasma cells of similar cells in other parts of the gut? Do the authors have data of their own showing that they are representative or can they refer to data of other investigators? If not, it should be indicated that this is a possible limitation of this study.

We concur completely with the reviewer's comment. This potential limitation was briefly evoked but not fully explained in the original manuscript. Due to the practical difficulties and ethic-regulatory constrains for obtaining gut biopsies elsewhere than the rectosigmoid tissue portion, we could not investigate B-cell and plasma cell repertoires in other parts of the GI tract. We have now modified the discussion part to present these points (page 17, lines 396-403): “Human gastro-intestinal tract tissues (i.e., jejunum, ileum and colon) display extensive sharing of B-cell clonal lineages (1), suggesting related B-cell repertoires across distinct intestinal locations. Moreover, B-cell memory sub-populations were shown to be similarly distributed in

duodenal, jejunal and rectal tissues of rhesus macaques infected with simian-human immunodeficiency virus (SHIV) (2). Nonetheless, we cannot ensure that the rectosigmoid-derived B-cell and plasma cell repertoires studied here are representative of those found elsewhere in the gut of ART-treated individuals leaving with HIV-1. Thus, whether HIV-1 infection also triggers the elicitation and expansion of polyreactive B-cell clones in the other anatomic sections of the intestine than the rectosigmoid segment remains to be investigated.”

5. Minor issues:

a) In line 46, the authors refer to IgD antibodies. What evidence is there of IgD antibodies in gut mucosa?

In the introduction, the mention of IgDs among mucosa-derived antibodies is made globally for all MALT sites including non-gut tissues such as the aerodigestive / respiratory tract mucosa-associated lymphoid structures where IgD are secreted (reviewed in (3)). The statement ends on highlighting the predominant role of IgAs in the GALT. We have slightly modified this sentence to avoid any confusion (page 3, lines 25-26). To our knowledge, so far, there are no formal evidence for the presence of IgD-secreting PC in the human gut. Thus, the GALT may not support IgD production. That said, it is apparently possible that circulating blood IgDs transudate into the gastro-intestinal tract lumen (4).

b) In line 347, insert word 'of' between 'one' and 'the'.

We apologize for the typo, which is now corrected accordingly.

Reviewer #3 (Remarks to the Author):

Planchais and colleagues investigated the effect of beginning ART during early vs chronic HIV infection on mucosal B cells and related population in peripheral blood. The authors make several important observations but a few are not sustained by the data presented. In addition, several observations are made without interpretation.

We thank the reviewer for her/his critical reading of our manuscript and for her/his suggestions.

Specific points:

1. Table S1 is confusing: the chronic (L) group is defined by starting ART in chronic phase of infection as referenced in another paper but several in the L group seem to belong to the E group based on when they started ART, some at the time of infection. Are the authors conflating time of diagnosis with time of infection?

We apologize for the confusion, and we have changed the heading of column G in table S1. In agreement with our previous collaborative study (5), and as mentioned in the material & methods section, people living with HIV-1 at the Fiebig stage VI correspond to the donors included in the IART group (even if some started with antiretrovirals as soon as HIV-infection was diagnosed, which explains why some people in the late group – namely L16, L26, L38 - have a duration before treatment = 0). Conversely, all patients in the early group had a Fiebig stage between I and V at the time of HIV diagnosis.

2. The authors state on line 114 that they carried out deep sequencing on matched colorectal and peripheral blood samples, indicating analysis of 7 in each group yet in the first part of the study, line 88, they state a total of 6 colorectal biopsies were performed (3 in each group). Which is it or were only 3 analyzed by single-cell and extended to 4 others for bulk analyses? This should be clarified. More importantly, L4 and L7 seem to have started ARVs shortly after infection which would categorize them as early. Again, is this a difference between time of diagnosis vs time of infection?

We apologize for the lack of precision. As requested, we have clarified this information on the use of samples for the different experimental investigations on page 6, lines 97-98. As explained above, we have clarified this point and we confirm that donors L4 and L7 were at a Fiebig stage VI when they started ART.

3. The mucosal IgG mutational load for VH does not appear to agree between single cell (Fig. 1E) and bulk (Fig. 1F) analyses yet authors state these were similar, line 117. The differences were modest yet highly significant for the latter but not the former. Does this indicate that single-cell analyses were not representative? This needs to be addressed.

We apologize for the lack of clarity. In both cases – single cell and HTS analyses - the mutation loads in mucosal IgG VH genes were decreased in IART compared to eART but only very slightly 18.7 vs 18.5 (single cells) and 22 vs 21.9 (HTS). Taking into consideration this small variation, the difference found statistically significant for the HTS but not for the single cell analysis resulted from the comparison of VH mutation loads from several millions of antibody sequences in the former while only about 150 could be used in the latter (numbers of analyzed sequences are indicated in the figure legend). According to the reviewer's request, we have rephrased the part on the single cell description for enhancing the coherence between both analyses (pages 5-6, lines 91-95).

4. Do authors have an explanation why the 2 HIV-specific antibodies would bind both p24 and gp140. Did this extend to gp120 and/or gp41?

We apologize for the confusion. The two antibodies described indeed cross-reacted strongly with p24 and gp140 but also with any other antigens tested including gp120, gp41 and non-HIV molecules. Therefore, it is very unlikely that those immunoglobulins correspond to *bona fide* high affinity antibodies matured against a specific HIV-1 antigen. Rather, we showed that these two antibodies are highly polyspecific binders making them react with all HIV-1 and non-HIV-1 antigens tested. Hence, we consider these as non-HIV specific antibodies. We have changed the description of the data in the revised manuscript to make this clearer (page 7, lines 122-124 and 128, and Figures S3B and S3C).

Further, how did the evidence of polyreactivity in Figures 2B and S3 break down by group and for intestinal mAbs reconstituted from people who are not HIV-infected?

According to the reviewer's comment, we have changed Figures 2B and S3 to better display the polyreactivity status of mAbs *per* group of ART-treated individuals. As these data reflect the polyreactivity potential of the mAbs tested (as precised page 7, lines 125-129), we observed higher binding to single antigens in IART compared to eART (see Figure below), as well as dual/triple binders to gp140, gp120 and gp41 (Figures 2B and S3). Unfortunately, as the tertiary-lymphoid structures in the colorectal tissues found in individuals leaving with HIV-1, and from which the memory B cells and plasma cells were isolated here, are not present in non-infected healthy individuals, such comparative study could not be made.

5. Is it common to see such differences as shown in Fig 2D and 2F between HEp-2 ELISA and IFA? What might be the explanation?

Over the past two decades, both assays – HEp-2 ELISA and IFA – have been used in parallel in hundreds of publications from several groups worldwide (e.g., E. Meffre, H. Wardemann, P.C. Wilson, M.C. Nussenzweig & others including ours) to evaluate the cross-reactivity to self-antigens by human monoclonals, and indeed it is not uncommon to observe such differences. The ELISA is based on the recognition of antigens in a HEp-2 cell lysate and is usually more sensitive in detecting polyreactive antibody binding than the IFA. On the other hand, the IFA, more stringent, also allows identifying fluorescence patterns associated with a given reactivity or cross-reactivity to certain cell structure components. For instance, in this study, one can appreciate in the corrected Table S2 (a column shift, for which we apologize, was responsible for erroneous marks in the HEp-2 IFA column), that most reactive mAbs by IFA with a clear fluorescence staining were negative in the ELISA and *vice et versa*. Therefore, HEp-2 ELISA and IFA are not completely equivalent but rather complementary for this type of analyses, which is the reason why we keep combining and presenting both analyses for accuracy.

6. Line 198-206 discusses one mAb, L2-168 that has a non-polyreactive specificity. Why did the authors feel this was of importance? This is already a long paper and this result does not appear to add any value to the overall message. The authors should consider removing this part.

Alternatively, one would say that L2-168 is the only antibody out of 200 for which we could unequivocally attribute a unique bacteria specificity without any polyreactivity. Since this observation was quite intriguing, we decided to thoroughly examine L2-168 binding properties to potentially identify its target. We found that

the antibody reacts specifically with the *B. subtilis* LTA, an observation that was initially discussed in the manuscript but probably not well enough. While we understand the reviewer's point of view, we would like to keep these data in the manuscript. Indeed, even if these details concern only one antibody, they have the benefit of showing that gut-derived antibodies can also lack polyreactivity and specifically target a given bacteria strain of the microbiota, and that the memory B cells expressing such specificity also circulate in the periphery. Moreover, considering the potent probiotic activity of *B. subtilis*, it is noteworthy to describe our finding of the presence of gut memory B cells targeting *B. subtilis* in a late treated individual. Therefore, we have extended the discussion point on this finding to better consider its potential relevance (page 15, lines 347-351).

7. Is it common for LPS reactivity to be driven by IgA1 over IgA2 when latter is the predominant source in the lower gastrointestinal tract?

This is an interesting point raised by the reviewer. Considering that plasma anti-LPS IgG and IgA antibodies are frequently detected in ART-treated patients (6), and that the vast majority (>90%) of circulating blood IgAs are IgA1 antibodies, it is not surprising to detect more anti-LPS IgA1 than IgA2 antibodies in the donors' sera. It is therefore plausible that an important proportion of these anti-LPS IgA1 arose from a peripheral rather gut-specific immune response. Yet, we cannot exclude that a substantial fraction of LPS-specific IgAs originate from the gut. Indeed, HIV-1 infection has been associated to a significant increase of gut IgA1-expressing plasma cells and -B cells over IgA2 levels (7), some of which may enter the long-lived and/or memory compartment and secrete IgA1 antibodies reaching the circulation. In addition, the very low quantity of serum IgA2 antibodies may render more difficult the detection of anti-LPS IgA2 as compared to IgA1. We have modified the discussion part on pages 16-17, lines 383-386 to detail more clearly these points.

8. From line 227: if striking, why not add Fig. S4E to main figure in lieu of Fig. 2D-F?

We thank the reviewer for this excellent suggestion. We have split the original Figure 4 into new Figures 4 and 5, and added the panel S4E at the end of the revised Figure 4. These changes provide a better balance with the results originally described in separate sections of the main text and allow to comprehend more clearly the data displayed in the figures.

9. Lines 237-248: Regarding elevated dIgA and SIgA, this may simply reflect overall increases in IgA that is associated with chronic HIV. Do these specific differences hold if corrected for differences in total IgA?

We apologize for the lack of clarity. To determine dIgA and SIgA levels, total circulating blood IgAs were purified using peptide M-coupled agarose beads as described in the Materials & Methods. ELISAs were performed using these purified serum IgA antibodies that were subsequently tested at 100 µg/ml and 3 consecutive 1:3 dilutions. This strategy allowed us to normalize for the total IgA concentration from tested samples ensuring the identification of an unbiased enrichment of dIgA and SIgA among total IgAs.

10. The most important point: the authors do not make a clear case for gut-related polyreactive blood CD27-IgA+ memory B cells in IART as stated in subtitle line 273 and Abstract line 30 and Discussion line 398. It is very unfortunate that the authors did not test more subjects to address questions posed starting line 275. It is understandable that colorectal biopsies are not routine procedures, but blood collection and phenotypic analyses are. The data shown in Figure 5B did not reach statistical significance, there was no increase in circulating IgA B cells that express the mucosal markers CD27-B7+.

We agree with the reviewer that performing these analyses on more samples would have strengthened our results. Unfortunately, we only had access to a limited number of PBMC samples that were bio-banked when receiving the colon biopsies from the same donors to be used for single cell sorting (3 *per* group were utilized for this matter, but we received biological materials (blood and colon biopsies) for 5 eART and 4 IART in total). Attempts in obtaining additional blood samples would considerably delay the suggested review experiments as it would require amending our clinical research protocol and obtaining the subsequent ethics and regulatory approvals; this is not feasible on a short term. Considering that this work is subjected to confirmed firm competition from other research labs, and that the data highlighted as different between groups were based on statistical analyses, we wish not to add more biological samples at this time. In this regard, and to directly answer the criticism on the results, we gently disagree with the reviewer's comment: it is true that we did not observe any statistical difference of the % of total IgA⁺CD27⁺b7⁺ cells between IART and eART (Figure 6B) – this is now clearly mentioned in the revised manuscript

page 12 line 288. Our claim described “*in subtitle line 273 and Abstract line 30 and Discussion line 398*” concerns the accumulation of polyreactive IgA⁺CD27⁺b7⁺ in IART, which is supported by the data shown in Figure 6E (p=0.016).

Figure 5C showing a higher proportion of gp140+ B cells among CD27-B7+ in IART is based on very few events and must be confirmed with more sampling.

In line with the studies from co-authors of this study (5) as well as from Moir et al (8), we observed that eART individuals exhibit a low frequency of circulating gp140-specific B cells, but much more than IART. Of note, the quantification of gp140-reactive B cells was performed by gating gp140⁺ cells on non-polyreactive B cells (Insulin-RNApol⁻LPS⁻) to avoid non-specific binding and ensure the specificity of the recognition; this rigorous approach decreased further the % of gp140⁺ B cells. Of note, the tSNE plots do not present all positive cells (normalized n cells *per* donors), which are better depicted on Figure 6D with the frequencies. Adding more samples for the analysis, which we unfortunately cannot presently do, would likely give a p value < 0.05 when comparing gp140⁺IgG⁺ B cells between groups. Nevertheless, despite the limited number of analyzed patients, these data are robust as confirmatory to already published works. Regarding the % of gp140⁺ among B-cell subsets, we have therefore moderated our statement page 12, lines 274-277.

The authors refer to polyreactive clones, beginning line 294, among circulating B cells based on binding to three proteins. It is not clear how the triple binding was determined based on the flow plots shown in Fig. S5A and it cannot be assumed that these proteins are binding to the BCR unless mAbs are cloned.

We apologize for the lack of clarity. As well understood by the reviewer, the polyreactivity of circulating B cells was determined by flow cytometry based on the simultaneous binding to LPS, Insulin and RNA pol α . These antigens were selected based on our published findings that they represent potent ligands bound by polyreactive monoclonal antibodies (9). The gating strategy is shown on Figure S5: we first identified positive B cells reacting with both RNA pol and insulin (Fig. S5A, top panels). We then evaluated the binding of these double positive B cells to LPS (Fig. S5A, bottom panels). The frequencies determined for the triple binders (LPS⁺, RNA pol⁺, insulin⁺ B cells), which we referred as polyreactive (polyR), are indicated with colored values (Fig. S5A, top panels). Details for this methodological approach are now given in the Material & Methods section (page 22, lines 520-522), and the figure S5 legend has been modified for a better understanding (page 41, lines 1137-1138). Flow cytometric binding has been previously used to determine the polyreactivity level of B cells (10, 11). More importantly, antigen capture by FACS and subsequent antibody cloning for a given molecule have generally confirmed the relationship between B-cell binding by flow cytometry and ELISA reactivity of the corresponding mAbs. When mAbs were found to be not specific, some studies could confirm that those bound the antigens through polyreactivity (9, 12). Hence, we are confident about the veracity of our data based on the definition of polyreactive B cells as those binding to the 3 unrelated molecules, especially as we also performed the analyses on either IgG- or IgA-expressing B cells in order to minimize low-affinity non-BCR binding that could occur through interactions with unknown cell surface molecules. While cloning and testing mAbs from blood B cells could be an option to confirm our results, we gently disagree with the reviewer's comment that our claim necessitates mAb cloning & testing, which we consider to be beyond the scope of the study.

The direct evidence of connection between peripheral and mucosal B cells comes from the sequences in Ig-HTS libraries from PBMCs that matched with mucosal mAb sequences. However, the polyreactive sequences were not broken down by IgA vs IgG which questions how much of the enrichment in IART is driven by IgA?

We thank the reviewer for his/her suggestion. We have modified the Figure 6F in order to display the proportion of IgA vs IgG and have modified the revised manuscript accordingly (page 13, line 295-296).

11. Overall, the authors show elements of increased polyreactivity of IgA mucosal origin and connection between blood and intestinal sequences but there is no clean link to a population in the peripheral blood. The authors need to address this lack of connection.

We thank the reviewer for this excellent suggestion. Unfortunately, the link between polyreactive single-cell sorted intestinal B cells and a specific blood B-cell sub-population of a given phenotype could not be addressed since blood B-cell sequences were obtained using Ig-HTS (Illumina technology) and not scRNA

seq. Hence, we only obtained IgH sequences of circulating blood IgA⁺ and IgG⁺ B cells but not of the global transcriptomic profiles that would have permitted B-cell immunophenotyping.

12. In the Discussion, line 322-324, the authors do not account for differences between their cohort and the one in reference 53: the latter was conducted on PLWH who were viremic at time of study. This statement should be modified accordingly.

We thank the reviewer for pointing out the different nature of the cohorts analyzed in the two studies. We have therefore modified the statement accordingly (page 14, line 309-310).

Minor points

1. IgG on surface of ASC can be low and difficult to evaluate without permeabilization. Were all ASC accounted for by an Ig isotype? If not, then it would suggest that those missing may be IgG and undercounted.

We thank the reviewer for pointing this out. In this study, all intestinal ASCs/plasmablasts were single-cell sorted based on surface IgA or IgG expression. Intracellular staining of ASCs would have required cell-fixation and -permeabilization steps, a process incompatible with our methodology as it leads to an extensive RNA degradation and inhibition of the RT-PCR amplifications. Considering the size, the rarity and preciousness of our sampled biopsies, we decided not to permeabilize the intestinal cells following isolation. We have mentioned the potential technical bias on page 5, line 74-75.

2. For readability, add LPS to Fig 3C.

We thank the reviewer for her/his suggestion. We have modified the figure accordingly.

3. Lines 139-141: it cannot be that surprising to have a very low frequency of HIV-specificity among the total memory B-cell repertoire for PLWH on ART. This low frequency does not necessarily suggest polyreactivity. The authors should not make such an assumption.

As explained above, we have clarified the interpretation on the antibody reactivity to both gp140 and p24 (Specific point 4).

1. A. L. Friedman, U. Hershberg, J. J. C. Thome, B. Zhang, T. Granot, D. Ren, D. J. Carpenter, W. Meng, G. W. Schwartz, N. Matsuoka, A. M. Rosenfeld, M. J. Shlomchik, D. L. Farber, H. Lerner, E. T. Luning Prak, An atlas of B-cell clonal distribution in the human body. *Nat. Biotechnol.* **35**, 879–884 (2017).
2. T. Demberg, V. Mohanram, D. Venzon, M. Robert-Guroff, Phenotypes and distribution of mucosal memory B-cell populations in the SIV/SHIV rhesus macaque model. *Clin. Immunol.* **153**, 264–76 (2014).
3. K. Chen, G. Magri, E. K. Grasset, A. Cerutti, Rethinking mucosal antibody responses: IgM, IgG and IgD join IgA. *Nat. Rev. Immunol.* **20**, 427–441 (2020).
4. K. Hjelt, C. H. Sørensen, O. H. Nielsen, P. A. Krasilnikoff, Concentrations of IgA, secretory IgA, IgM, secretory IgM, IgD, and IgG in the upper jejunum of children without gastrointestinal disorders. *J. Pediatr. Gastroenterol. Nutr.* **7**, 867–71 (1988).
5. C. Planchais, L. Hocqueloux, C. Ibanez, S. Gallien, C. Copie, M. Surenaud, A. Kök, V. Lorin, M. Fusaro, M.-H. Delfau-Larue, L. Lefrou, T. Prazuck, M. Lévy, N. Seddiki, J.-D. Lelièvre, H. Mouquet, Y. Lévy, S. Hüe, Early Antiretroviral Therapy Preserves Functional Follicular Helper T and HIV-Specific B Cells in the Gut Mucosa of HIV-1–Infected Individuals. *J. Immunol.* **200**, 3519–3529 (2018).
6. A. Lim, A. Amini, L. J. D’Orsogna, R. Rajasuriar, M. Kramski, S. R. Lewin, D. F. Purcell, P. Price, M. A. French, Antibody and B-cell responses may control circulating lipopolysaccharide in patients with HIV infection. *AIDS.* **25**, 1379–83 (2011).

7. S. T. Jones, K. Guo, E. H. Cooper, S. M. Dillon, C. Wood, D. H. Nguyen, G. Shen, B. S. Barrett, D. N. Frank, M. Kroehl, E. N. Janoff, K. Kechris, C. C. Wilson, M. L. Santiago, Altered Immunoglobulin Repertoire and Decreased IgA Somatic Hypermutation in the Gut during Chronic HIV-1 Infection. *J. Virol.* **96**, e0097622 (2022).
8. L. Kardava, S. Moir, N. Shah, W. Wang, R. Wilson, C. M. Buckner, B. H. Santich, L. J. Y. Kim, E. E. Spurlin, A. K. Nelson, A. K. Wheatley, C. J. Harvey, A. B. McDermott, K. W. Wucherpfennig, T.-W. Chun, J. S. Tsang, Y. Li, A. S. Fauci, Abnormal B cell memory subsets dominate HIV-specific responses in infected individuals. *J. Clin. Invest.* **124**, 3252–3262 (2014).
9. C. Planchais, A. Kök, A. Kanyavuz, V. Lorin, T. Bruel, F. Guivel-Benhassine, T. Rollenske, J. Prigent, T. Hieu, T. Prazuck, L. Lefrou, H. Wardemann, O. Schwartz, J. D. Dimitrov, L. Hocqueloux, H. Mouquet, HIV-1 Envelope Recognition by Polyreactive and Cross-Reactive Intestinal B Cells. *Cell Rep.* **27**, 572-585.e7 (2019).
10. Z.-H. Zhou, Y. Zhang, Y.-F. Hu, L. M. Wahl, J. O. Cisar, A. L. Notkins, The broad antibacterial activity of the natural antibody repertoire is due to polyreactive antibodies. *Cell Host Microbe.* **1**, 51–61 (2007).
11. Z. H. Zhou, A. G. Tzioufas, A. L. Notkins, Properties and function of polyreactive antibodies and polyreactive antigen-binding B cells. *J. Autoimmun.* **29**, 219–228 (2007).
12. V. Hehle, M. Beretta, M. Bourguin, M. Ait-Goughoulte, C. Planchais, S. Morisse, B. Vesin, V. Lorin, T. Hieu, A. Stauffer, O. Fiquet, J. D. Dimitrov, M. L. Michel, M. N. Ungeheuer, C. Sureau, S. Pol, J. P. Di Santo, H. Strick-Marchand, N. Pelletier, H. Mouquet, Potent human broadly neutralizing antibodies to hepatitis B virus from natural controllers. *J. Exp. Med.* **217** (2020), doi:10.1084/jem.20200840.

REVIEWERS' COMMENTS

Reviewer #2 (Remarks to the Author):

The authors have responded carefully and effectively to the comments made by reviewers. I am satisfied with the responses to my comments.

Reviewer #3 (Remarks to the Author):

Overall, the authors have made substantial improvements to their manuscript. However, several analyses and conclusions were based on a limited number of samples. The authors argue (in point # 10) that it would take too long to increase the N and could lose to competitors. This is unfortunate but should not be a reason for a pass. At the very least, the authors should explicitly acknowledge that their study has limitations due to limiting sampling for cell-based assays and that corresponding conclusions need to be substantiated by larger studies.

Specific points that should be addressed.

Point #1:

Thank you for clarifying time of diagnosis. However, the only mention of Fiebig stage in the manuscript is for the chronic group, not early group. Table S1 would be more helpful if it included a column with the Fiebig stage of each participant.

Point #3:

As far as this reviewer can tell (none of the page and line numbers match between letter and manuscript), one word was changed in explaining the difference between HGS and single-cell sequencing. The HGS data are much more robust than single cell data, which are based on far fewer sequences. Per the author's own words, the HGS data for mucosal IgA showed a "modest" difference. For the single cell data, the authors use the term "strikingly" and while the margin was greater than for HGS, the caveat is that the single cell data were based on far fewer sequences. In the letter the authors state single cell analyses were based on ~150 sequences and refer to details in the figure legend. The legend indicates n=143 for eART and n=138 for IART; however, this appears to be for combined IgG and IgA. This means the single cell data presented for mucosal IgG and IgA were based on less than 100 sequences each. The statements and conclusions regarding mutational loads should be modified to reflect these limitations.

We thank the reviewers for their positive feedback. We wish to thank them once again for their insightful comments, which helped us improving the manuscript.

Reviewer #2 (Remarks to the Author):

The authors have responded carefully and effectively to the comments made by reviewers. I am satisfied with the responses to my comments.

Reviewer #3 (Remarks to the Author):

Overall, the authors have made substantial improvements to their manuscript. However, several analyses and conclusions were based on a limited number of samples. The authors argue (in point # 10) that it would take too long to increase the N and could lose to competitors. This is unfortunate but should not be a reason for a pass. At the very least, the authors should explicitly acknowledge that their study has limitations due to limiting sampling for cell-based assays and that corresponding conclusions need to be substantiated by larger studies.

We thank the reviewer for her/his suggestions. Accordingly, we have modified the discussion to include these limitations, page 18, lines 422-424.

Specific points that should be addressed.

Point #1:

Thank you for clarifying time of diagnosis. However, the only mention of Fiebig stage in the manuscript is for the chronic group, not early group. Table S1 would be more helpful if it included a column with the Fiebig stage of each participant.

We apologize for the missing information. According to the reviewer's request, we have added a column in Table S1 indicating for each donor their Fiebig stage.

Point #3:

As far as this reviewer can tell (none of the page and line numbers match between letter and manuscript), one word was changed in explaining the difference between HGS and single-cell sequencing. The HGS data are much more robust than single cell data, which are based on far fewer sequences. Per the author's own words, the HGS data for mucosal IgA showed a "modest" difference. For the single cell data, the authors use the term "strikingly" and while the margin was greater than for HGS, the caveat is that the single cell data were based on far fewer sequences. In the letter the authors state single cell analyses were based on ~150 sequences and refer to details in the figure legend. The legend indicates n=143 for eART and n=138 for IART; however, this appears to be for combined IgG and IgA. This means the single cell data presented for mucosal IgG and IgA were based on less than 100 sequences each. The statements and conclusions regarding mutational loads should be modified to reflect these limitations.

We apologize for the apparent discrepancies between pages/lines # in the letter and the revised manuscript.

Despite that the single memory B-cell data, looking in this case at SHM levels, are based on much less sequences than for the HTS analyses, once again the difference was highly statistically significant for IgAs ($p=0.009$) and to our interpretation, as robust if not more than the results obtained by Ig-HTS. Apart from the well-known nucleotide-specific biases associated with HTS library preparation, sequencing process and analyses that we have minimized as much as possible with our *in vitro* and *in silico* methodological pipeline, the nature of the starting material is key to consider: for the single cell, the data being discussed here were obtained from the analysis of *bona fide* gut memory B cells well phenotyped by flow cytometry, while Ig-HTS data were generated from more heterogenous libraries as containing transcripts of both B cells (naïve, memory...) and plasmablasts/plasma cells. Considering that plasmablasts/plasma cells were as abundant as memory B cells in human rectosigmoid biopsies (see Supplementary Fig. 1b), and that Ig transcript levels in plasmablasts/plasma cells are several orders of magnitude over those of memory B cells, it is highly likely that the corresponding datasets are greatly enriched in plasmablast/plasma cell-derived sequences. Since single ASC data (in Supplementary Figure 2g) showed reverse tendencies that may counterbalance those found for memory B cells (less SHM in IgA and more in IgG for IART compared to eART), it is therefore not surprising to observe a less

pronounced difference when looking at the SHM loads in IgA by Ig-HTS between groups. We have modified the results section to clarify this point - page 6 (underlined sentences).